# The value of remote marine aerosol measurements for constraining radiative forcing uncertainty

Leighton A. Regayre[1], Julia Schmale[2,3], Jill S. Johnson[1], Christian Tatzelt[4], Andrea Baccarini[2], Silvia Henning[4], Masaru Yoshioka[1], Frank Stratmann[4], Martin Gysel-Beer[2], Daniel P. Grosvenor[1,5] and Ken S. Carslaw[1]

[1]Institute for Climate and Atmospheric Science, School of Earth and Environment, University of Leeds, Leeds, LS2 9JT, UK
[2]Paul Scherrer Institute, Laboratory of Atmospheric Chemistry, Villigen, Switzerland
[3]École Polytechnique Fédéderale de Lausanne, Lausanne, Switzerland
[4]Leibniz Institute for Tropospheric Research, Leipzig, Germany
[5]National Centre for Atmospheric Science, Leeds, UK
*Correspondence to*: Leighton Regayre (L.A.Regayre@leeds.ac.uk)
*Correspondence related to measurements to:* Julia Schmale (julia.schmale@psi.ch)

**Abstract.** Aerosol measurements over the Southern Ocean are used to constrain aerosol-cloud interaction radiative forcing ($RF_{aci}$) uncertainty in a global climate model. Forcing uncertainty is quantified using one million climate model variants that sample the uncertainty in nearly 30 model parameters. Measurements of cloud condensation nuclei and other aerosol properties from an Antarctic circumnavigation expedition strongly constrain natural aerosol emissions: default sea spray emissions need to be increased by around a factor of 3 to be consistent with measurements. Forcing uncertainty is reduced by around 7% using this set of several hundred measurements, which is comparable to the 8% reduction achieved using a diverse and extensive set of over 9000 predominantly Northern Hemisphere measurements. When Southern Ocean and Northern Hemisphere measurements are combined, uncertainty in $RF_{aci}$ is reduced by 21% and the strongest 20% of forcing values are ruled out as implausible. In this combined constraint, observationally plausible $RF_{aci}$ is around 0.17 W m$^{-2}$ weaker (less negative) with 95% credible values ranging from -2.51 to -1.17 W m$^{-2}$ (standard deviation -2.18 to -1.46 W m$^{-2}$). The Southern Ocean and Northern Hemisphere measurement datasets are complementary because they constrain different processes. These results highlight the value of remote marine aerosol measurements.

## 1   Introduction

The uncertainty in the magnitude of the effective radiative forcing caused by aerosol-cloud interactions ($ERF_{aci}$) due to changing emissions over the industrial period is around twice that for $CO_2$ (Stocker et al., 2013). It is essential to reduce this uncertainty if global climate models are to be used to robustly predict near-term changes in climate (Andreae et al., 2005, Myhre et al., 2013, Collins et al., 2013, Tett et al., 2013, Seinfeld et al., 2016).

Aerosol forcing uncertainty has persisted in climate models since the 1990s partly because there are no measurements covering the industrial period that can be used to directly constrain simulations of long-term changes in aerosol and cloud properties (Gryspeerdt et al., 2017; McCoy et al., 2017). Estimates of aerosol forcing over the industrial period therefore rely on models that have been evaluated against measurements made in the present-day atmosphere. However, it is known that the aerosol forcing (in particular the component caused by aerosol-cloud interactions) depends sensitively on the state of aerosols in the pre-industrial period (Carslaw et al., 2013; Wilcox et al. 2015) when natural aerosols were dominant (Carslaw et al., 2017). Observations of natural aerosols in the present-day atmosphere are therefore expected to help constrain the simulated forcing unless there have been significant changes in natural aerosol processes over the industrial period, for which there is little evidence (Carslaw et al., 2010).

In this paper we address the questions: i) To what extent can measurements of aerosols in pristine (natural) environments help to constrain model simulations and thereby reduce the large uncertainty in aerosol forcing?

ii) What is the relative importance of measurements in remote and polluted environments for constraining the forcing uncertainty? It is known that the abundance of natural aerosols affects the magnitude of forcing in a model (Spracklen and Rap, 2013; Carslaw et al., 2013). However, to assess the effect on the *uncertainty* in forcing it is necessary to explore how the spread of predictions of a set of models changes when constrained by measurements. The 5th Coupled Model Intercomparison Project is inadequate for this purpose because of insufficient aerosol diagnostics (Wilcox et al., 2015). Here we use large perturbed parameter ensembles (PPEs) of the UK Hadley Centre General Environment Model HadGEM3 (Hewitt et al, 2011). The PPEs were created by systematically perturbing numerous model parameters related to natural and anthropogenic emissions and physical processes (Yoshioka et al., 2019). The simulated aerosol forcings have uncertainty ranges that exceed those of multi-model ensembles (Yoshioka et al., 2019; Johnson et al., 2019). Instantaneous radiative forcing (RF) is quantified using the 26-parameter AER PPE in which just aerosol-related parameters were varied, and the effective radiative forcing (ERF) is quantified using the 27-parameter AER-ATM PPE in which aerosol and physical atmosphere parameters were varied (Yoshioka et al., 2019). We use these PPEs to quantify how the constraint provided by pristine aerosol measurements affects the spread of aerosol forcings simulated by the ensembles.

Previous analysis of HadGEM3 PPEs showed that measurements of the present-day atmosphere in regions affected by anthropogenic emissions help to constrain the uncertainty in aerosol-radiation interaction forcing ($RF_{ari}$) but not the component due to aerosol-cloud interactions ($RF_{aci}$). For example, Regayre et al. (2018) showed that top-of-the-atmosphere shortwave radiation flux measurements reduce $ERF_{aci}$ uncertainty by only around 10%, despite the fluxes in the present-day and early-industrial environments sharing multiple causes of uncertainty. Johnson et al. (2019) showed that a much larger dataset of over 9000 (predominantly Northern Hemisphere) aerosol measurements reduced the uncertainty in global, annual mean aerosol $RF_{ari}$ (neglecting rapid adjustments) by 35%, but $RF_{aci}$ uncertainty by only around 7%. These measurements reduce the uncertainty in a small number of parameters related to anthropogenic emissions and aerosol processing in polluted environments. However, important causes of uncertainty in $RF_{aci}$, such as natural aerosol emission fluxes, were largely unconstrained.

The Southern Ocean is one of the few regions on Earth (along with some boreal forests) in which the same processes are expected to affect cloud-active aerosol concentrations in the present-day and early-industrial atmospheres (Hamilton et al., 2014). In this study we make use of aerosol measurements from the Antarctic Circumnavigation Expedition: Study of Preindustrial-like Aerosols and Their Climate Effects (ACE-SPACE) campaign (Schmale et al., 2019). They offer a unique opportunity to constrain the early-industrial aspects of aerosol forcing uncertainty because the Southern Ocean is a source of natural aerosols that are relevant at the global scale and remains largely unaffected by anthropogenic aerosol and precursor emissions.

We use near-surface measurements of cloud condensation nuclei concentrations at 0.2% and 1.0% supersaturations ($CCN_{0.2}$ and $CCN_{1.0}$; Tatzelt et al., 2019), as well as mass concentrations of non-sea-salt sulfate particles with dry aerodynamic diameters less than 10 μm and number concentrations of particles with dry aerodynamic diameter larger than 700 nm ($N_{700}$; corresponds to volume equivalent diameter larger than around 500 to 570 nm; Schmale et al., 2019a). The measurements are compared to output from 1 million variants of the HadGEM3 model that sample combinations of parameter settings in the model. These model variants are used to represent aerosol forcing uncertainty in our model using probability density functions (pdfs) and were generated by sampling from Gaussian Process emulators that were trained on the PPE model outputs (see SI Methods). Model variants that were judged to be observationally implausible against the measurements were rejected, resulting in a set of plausible variants from which the uncertainty in aerosol forcing could be computed (see SI Methods). In the results shown below, we retained approximately 3% of model variants (following Johnson et al., 2019) that best match all four measured aerosol properties.

## 2   Results

Fig. 1 shows the $CCN_{0.2}$ mean and standard deviation from the unconstrained and constrained model variants to exemplify the effect of constraint on model output. The mean concentrations in the unconstrained sample are much smaller than measured concentrations. However, the range of $CCN_{0.2}$ values in the unconstrained sample spans the measurements in most locations (Fig. 1b). The measurement constraint increases $CCN_{0.2}$ concentrations (more than double the unconstrained mean in many locations; Fig. 1c) and greatly reduces the $CCN_{0.2}$ uncertainty (by more than half everywhere to less than 50 cm$^{-3}$; Fig. 1d).

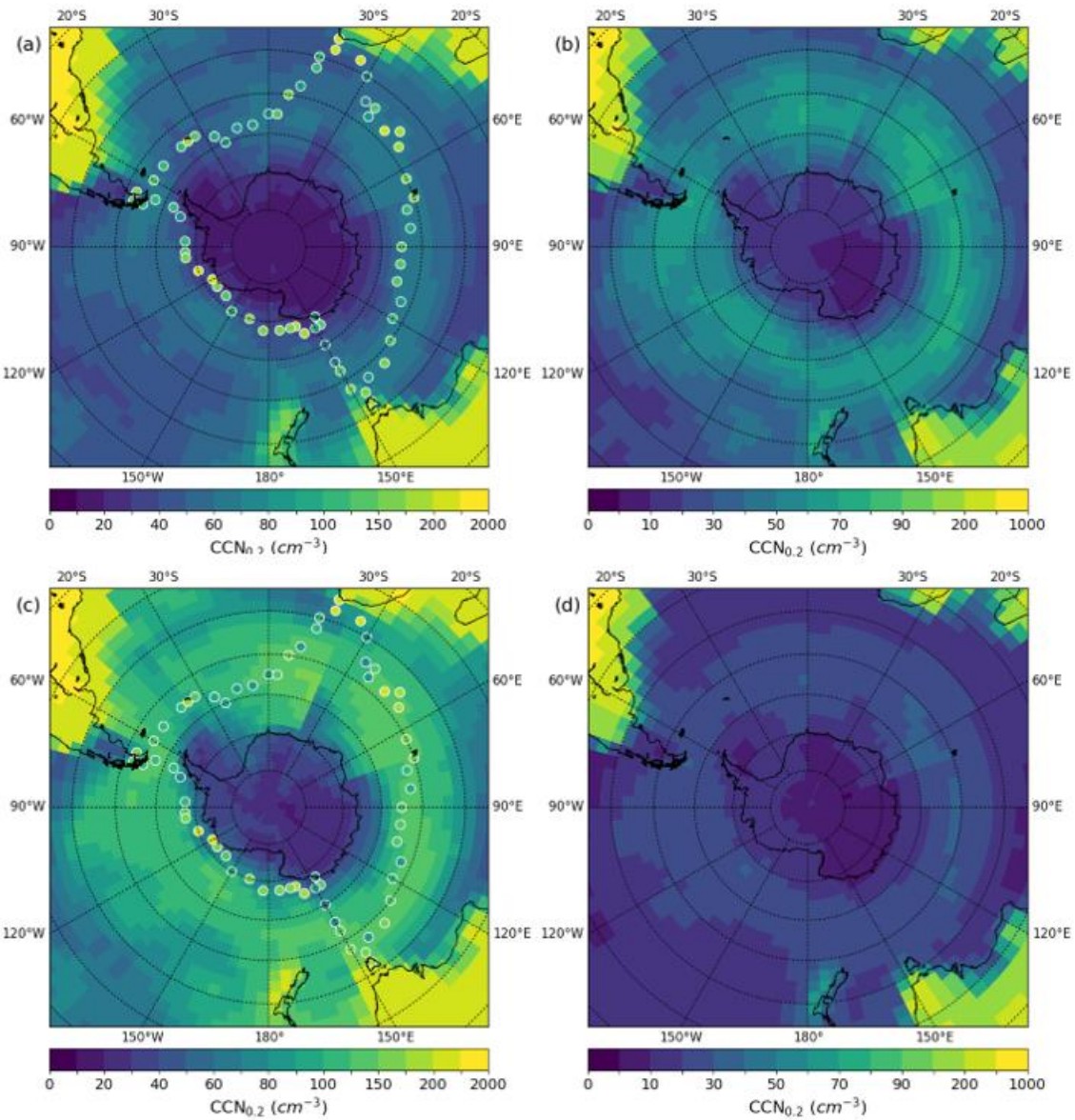


Fig. 1. a,c) Mean and b,d) standard deviation of CCN$_{0.2}$ concentrations from the a,b) unconstrained sample and c,d) the
sample constrained using concentration measurements of CCN$_{0.2}$, CCN$_{1.0}$, non-sea-salt sulfate and particles with dry
aerodynamic diameters larger than 700 nm. Measured CCN$_{0.2}$ values are plotted as dots. Means and standard deviations were
calculated using samples taken from emulators trained using monthly mean values. December to March sample values were
combined based on longitudinal agreement with measurements.

Fig. 2 shows pdfs of the output from the model for the four variables used as constraints, calculated as means
over the locations where measurements were taken. The constraint reduces the uncertainty in all measurement
types (narrower pdfs) and the central tendency of the pdfs is closer to the regional mean of measurements after
constraint. Rejecting around 97% of model variants as implausible compared to measurements greatly improves
the model-measurement comparison.

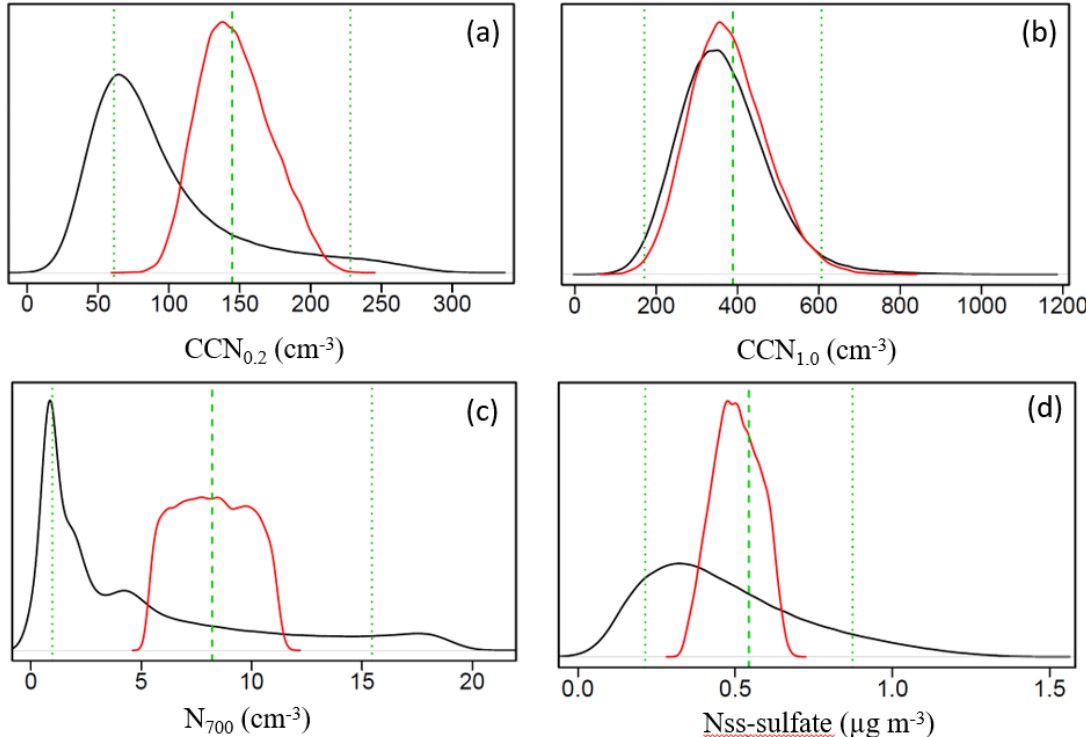

Fig. 2. Unconstrained (black) and observationally constrained (red) pdfs of aerosol properties: a) $CCN_{0.2}$, b) $CCN_{1.0}$, c) $N_{700}$ and d) aerosol sulfate. The pdfs were calculated at locations where measurements were used for constraint across the months December to March. Densities for each sample of model variants are scaled so that the area under the curve integrates to one. The green dashed line shows the median of the measurements and the dotted green lines show the approximate uncertainty ranges due to multiple model-measurement comparison uncertainties that were accounted for in the constraint (See SI Methods).

After constraint, the remaining model variants inhabit specific parts of the 26-dimensional parameter uncertainty space used to quantify the model uncertainty. We explore the effect of constraints on parameter values using 1-dimensional marginal probability distributions (described in detail in Johnson et al., 2019) – see Fig. 3 and Fig. S2 for equivalent AER-ATM results. The magnitude of the marginal probability distribution after constraint reflects the number of ways in which a particular value of a parameter can be combined with settings of all the other parameters to produce an observationally plausible model. The white space in the marginal pdfs shows where parameter value density has decreased.

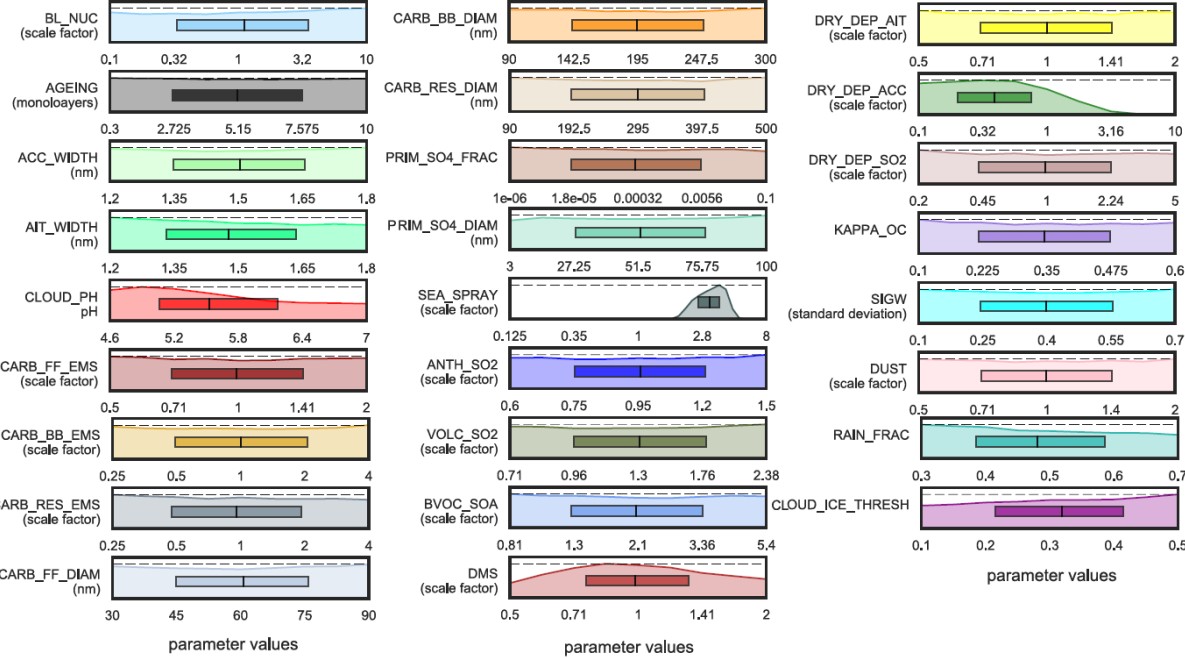

Fig. 3. Marginal probability distributions for the 26 aerosol parameters after constraint using ACE-SPACE measurements. The density of parameter values in the unconstrained sample are shown as horizontal dashed lines (uniform sampling over the parameter space). Densities of constrained samples are shown in colour and are scaled so that the maximum densities in the constrained and unconstrained samples are aligned. The 25th, 50th and 75th percentiles of each marginal distribution are shown in the central boxes. Parameter values on the x-axes correspond to values used in the model (Yoshioka et al., 2019, table S3).

The relative simplicity of aerosol emissions and processes over the Southern Ocean (compared to polluted continental regions) means that measurements can be used to tightly constrain uncertainty in the associated parameters. Two parameters (sea spray emissions and dry deposition velocity) are tightly constrained such that some parameter values are ruled out as implausible even when combined with uncertainties in all other parameters. Several other parameters (related to cloud droplet pH, dimethylsulfide (DMS) emissions and wet deposition) are more modestly constrained. These joint constraints (see also Fig. S3) suggest the model-measurement comparison is improved when aerosol number concentrations and mass are relatively high.

Sea spray emissions are tightly constrained to be around 3 times larger than the default model value. Observationally plausible values of the sea spray scaling parameter range from around 1.6 to 5.1 and all other values (including the default emission calculated in the model) are ruled out as implausible. This suggests that sea spray emissions in our model need to be significantly higher than those calculated using the wind speed dependent Gong (2003) parametrisation in the Southern Hemisphere summer. The higher flux is consistent with Revell et al. (2019), who showed that a more recent version of our model simulates cloud droplet concentrations and aerosol optical depth values that are lower than observed over the Southern Ocean in the Southern Hemisphere summer. However, in the Southern Hemisphere winter Revell et al., (2019) simulated higher aerosol optical depths than observed, which they corrected by reducing the dependence of sea spray emissions on wind speed. Hence, our constraint on sea spray emission fluxes may only be appropriate for Southern Hemisphere summer when wind speeds are relatively low. We do not make any assumptions about the composition of these additional summertime sea spray particles. They may be rich in organic material as proposed by Gantt et al. (2011) which would alter the CCN activity of emitted particles. However, the consistency of constraint of $CCN_{0.2}$ and $N_{700}$ towards higher values (Fig. 2, table S3) implies that a general scaling of the existing sea spray flux is consistent with the measurements from December to April, without the need for an additional source of fine-mode, organic-rich particles.

The dry deposition velocity of accumulation mode aerosols (Dry_Dep_Acc) has an 84% likelihood of being lower than the default model value after applying the constraint. Furthermore, deposition velocities larger than around 3 times the default value are effectively ruled out. This constraint is consistent with the higher aerosol concentrations implied by constraint of the sea spray emission parameter.

Other parameters are more modestly constrained. The constraint on the aerosol precursor DMS emission flux
scale factor is two-sided, reducing the credible range of DMS emission scalings from 0.5 to 2.0 down to 0.54 to
1.9. This constraint suggests the default surface sea water concentration (Kettle and Andreae, 2000) and
emission parameterisation (Nightingale, et al., 2000) are consistent with measurements (including aerosol
sulfate) and do not benefit from being scaled. Furthermore, ACE-SPACE measurements are consistent with
less-efficient aerosol scavenging (55% likelihood of Rain_Frac, the parameter that controls the fractional area of
the cloudy part of model grid boxes where rain occurs, being below the unconstrained median value 0.5) and
less aqueous phase sulfate production (pH of cloud droplets has a 62% likelihood of being lower than the
unconstrained median value). These combined constraints suggest, in agreement with sea spray and deposition
parameter constraints, higher aerosol number and mass concentrations are consistent with measurements.
The effects of measurement constraint on pdfs of $RF_{aci}$ and $ERF_{aci}$ are shown in Fig. 4. Removing implausible
model variants has reduced the uncertainty in several parameters including natural aerosol emission fluxes,
which translates into a reduction in $RF_{aci}$ uncertainty (Carslaw et al., 2013). The measurement constraints have
two important effects on aerosol forcing. Firstly, the magnitude of median $RF_{aci}$ weakens from -1.99 W m$^{-2}$ to -
1.88 W m$^{-2}$ (-1.64 to -1.49 W m$^{-2}$ for $ERF_{aci}$). A weaker forcing is consistent with higher natural aerosol
emissions, increased aerosol load and higher cloud droplet number concentrations in the early-industrial period.
Table 1 shows that our constraint on natural emission parameters also constrains Southern Ocean cloud droplet
number concentrations towards higher values, reducing the credible interval by around 50% and bringing mean
values into closer agreement with MODerate Imaging Spectroradiometer (MODIS; Salomonson et al., 1989)
instrument data (note that droplet number concentrations were not used to constrain the model). Thus, we
conclude that the constraint on aerosol forcing towards weaker values is a genuine constraint and not the result
of an arbitrary tuning. Secondly, the constrained forcing pdfs are approximately symmetric but have shorter tails
(lower kurtosis). This suggests the constraints are selectively ruling out model variants that are outliers. The
95% credible range of $RF_{aci}$ values is reduced by around 9% (from -2.84 to -1.15 W m$^{-2}$ down to -2.64 to -1.10
W m$^{-2}$) and around 9% for $ERF_{aci}$ (from -2.69 to -0.62 W m$^{-2}$ down to -2.43 to -0.54 W m$^{-2}$). The consistency of
forcing constraint across two distinct PPEs suggests the results are insensitive to differences in meteorology,
parameters perturbed in the PPEs, and the inclusion of rapid atmospheric adjustments. These results are also
insensitive to additional constraint to ensure energy balance at the top of the atmosphere (Fig. S5).

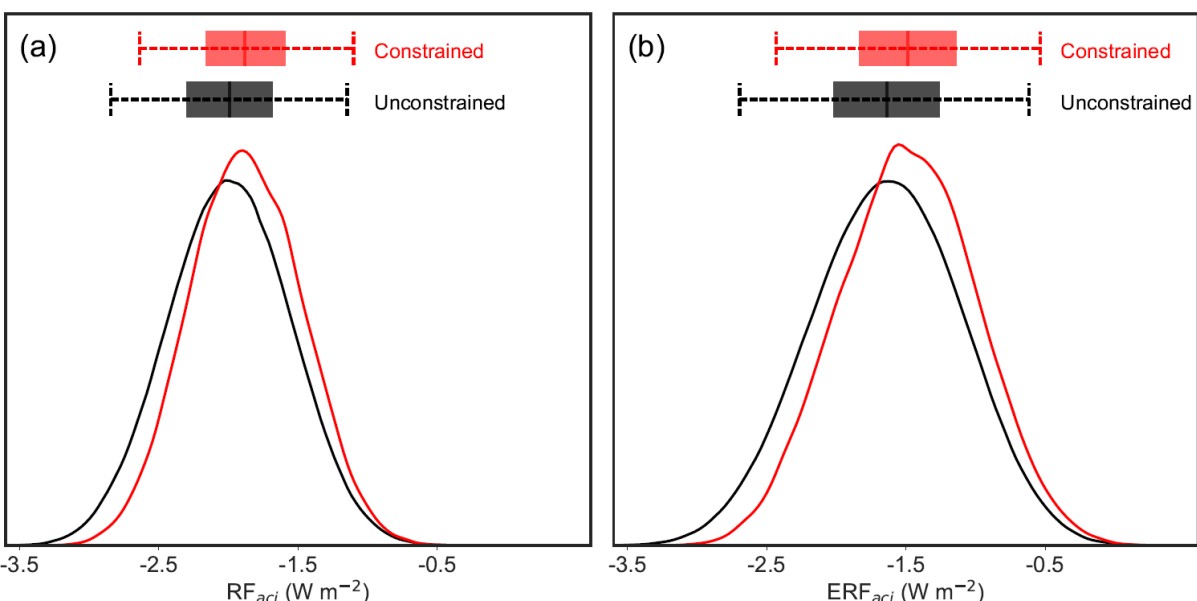

Fig. 4. Probability distributions of a) $RF_{aci}$ and b) $ERF_{aci}$. The distributions of the unconstrained sample of one million model
variants from statistical emulators of each PPE are in black. Red lines show the distributions after constraint using ACE-
SPACE measurements (around 3% of the unconstrained sample). The 25th, 50th and 75th percentiles of each sample are
shown as shaded boxes and dashed lines span the 2.5th and 97.5th percentiles.
Table 1. Annual and monthly mean cloud drop number concentrations over the Southern Ocean (over the region between
50ºS and 60ºS at around 1km altitude above sea level) in the original unconstrained sample and the sample of model variants
constrained to ACESPACE campaign measurements. Mean values and 95% credible interval values are shown for each
sample, with interquartile ranges in brackets. For comparison, we show cloud drop concentrations calculated from MODIS
instrument data following Grosvenor et al., (2018) for the year 2008 (SI Methods: Measurements).

|  | Annual | December | January | February | March | April |
|---|---|---|---|---|---|---|
| MODIS (cm$^{-3}$) | 73 | 89 | 91 | 90 | 82 | 63 |
| Unconstrained mean (cm$^{-3}$) | 38 | 39 | 39 | 41 | 42 | 39 |
| Unconstrained credible interval (cm$^{-3}$) | 7-125 (112) | 8-115 (103) | 8-117 (109) | 7-122 (115) | 7-129 (122) | 7-118 (111) |
| Constrained mean (cm$^{-3}$) | 66 | 67 | 69 | 72 | 76 | 70 |
| Constrained credible interval (cm$^{-3}$) | 41-96 (55) | 43-96 (53) | 44-99 (55) | 45-105 (60) | 47-111 (64) | 44-101 (57) |

Johnson et al. (2019) reduced the global, annual mean RF$_{aci}$ uncertainty by constraining multiple anthropogenic
emission and model process parameters (as well as some natural aerosol parameters) using over 9000
predominantly Northern Hemisphere measurements of aerosol optical depth, PM$_{2.5}$, particle number
concentrations and mass concentrations of organic carbon and sulfate. We used the same methodology as
Johnson et al. (2019) to rule out implausible model variants from the same original sample of one million model
variants, so we can readily combine these constraints. Around 700 model variants (0.07%) are observationally
plausible in both the Southern Ocean (ACE-SPACE) and Johnson et al. (2019) constraints. Although this is a
relatively small percentage of the original sample, 700 observationally-plausible model variants is far more than
are typically used to quantify model uncertainty or multi-model diversity (e.g. around 30 for CMIP6). The
marginal parameter pdfs from this 700-member sample are shown in Fig. 5. Because Johnson et al. (2019)
studied only the AER PPE (from which RF$_{aci}$ can be computed) we are unable to explore the effect of the
combined constraint on ERF$_{aci}$.

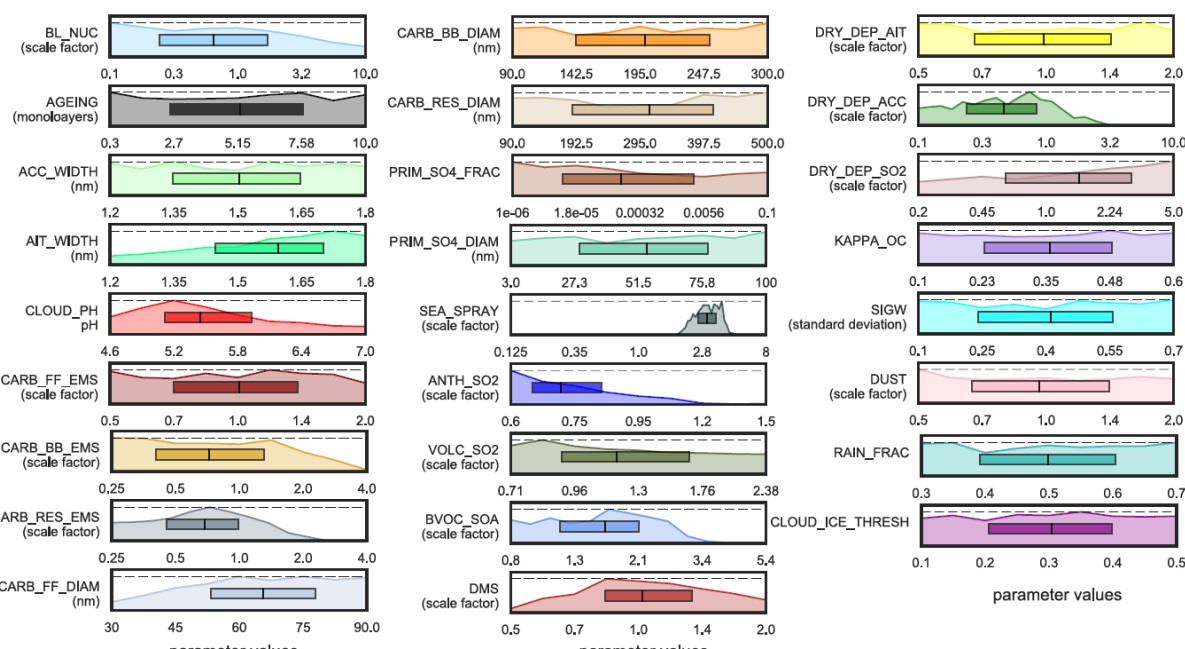

Fig. 5. Marginal probability distributions for the 26 aerosol parameters after constraint using around 250 Southern Ocean
measurements and more than 9000 aerosol measurements in Johnson et al. (2019). Plotting features of this figure are
identical to Fig. 3.
The two measurement datasets constrain distinct groups of parameters. There are a few cases where the same
parameters are constrained by both datasets and in these cases the parameter values are constrained consistently
(e.g. cloud droplet pH) or more strongly through ACE-SPACE (e.g. sea spray emissions). The complementary
nature of these constraints means that the combined constraint marginal parameter pdfs (Fig. 5) are remarkably
similar to those in our Fig. 3e (for sea spray and DMS emission fluxes, as well as deposition and pH parameters)
and in figure 6 of Johnson et al. (2019) for other parameters.

The Johnson et al. (2019) constraint reduced the RF_aci uncertainty by around 6% and our ACE-SPACE
measurement constraint reduced the uncertainty by around 9%. However, the RF_aci uncertainty is reduced by
around 21% (Fig. 6a) after applying both constraints, meaning the combined constraint is stronger than the sum
of individual constraints.

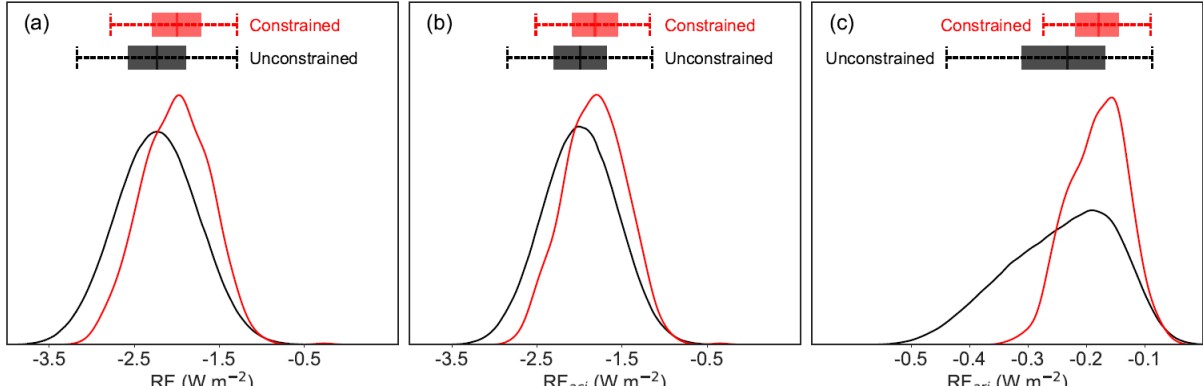

Fig. 6. Probability distributions of a) RF, b) RF_aci and c) RF_ari from the unconstrained (black line) and constrained (red line)
samples of model variants. The constrained sample includes model variants that agree with our ACE-SPACE measurement
constraint and the Johnson et al. (2019) constraint. Plotting features are identical to Fig. 4.
The Johnson et al. (2019) constraint strengthened the RF_aci by around 0.3 W m$^{-2}$ (more negative) because the
largest sea spray emission flux scaling and largest new particle formation rates were ruled out (Fig. 6 in Johnson
et al., 2019). Our ACE-SPACE constraint rules out the same large sea spray emission fluxes, but also rules out
all emission flux scale factors lower than around 1.6 (Fig. 3), which increases the baseline aerosol concentration
in the early-industrial atmosphere. The ACE-SPACE measurements also constrain several other parameters that
collectively weaken the median RF_aci by around 0.18 W m$^{-2}$. Therefore, using the combined measurement
dataset, the strongest RF_aci values have been ruled out as implausible and the credible range of observationally
plausible RF_aci values is reduced to around -2.51 to -1.17 W m$^{-2}$ (-2.18 to -1.46 W m$^{-2}$, when using one standard
deviation to quantify the uncertainty). Uncertainty in RF_ari is reduced by around 48% with observationally
plausible values ranging from -0.27 to -0.09 W m$^{-2}$ (-0.23 to -0.13 W m$^{-2}$, when using one standard deviation),
because the strongest RF_ari values are ruled out as observationally implausible.
**3    Discussion**
Our results show, as hypothesised from previous sensitivity analyses, that remote marine measurements are
valuable for constraining the natural aerosol state of the atmosphere (Carslaw et al., 2013; Regayre et al., 2014;
Regayre et al., 2018). Remote marine aerosol measurements provide new information about plausible model
behaviour because they are closely related to model emissions and processes that measurements in polluted
environments do not constrain.
For the first time we have achieved a meaningful reduction of 21% in the RF_aci uncertainty by constraining the
aerosol properties in the model. The reduction in forcing uncertainty can still be improved by considering the
following: Firstly, using measurements of cloud properties and cloud-aerosol relations, as well as measurements
associated with primary sulfate and carbonaceous particle emission sizes, could constrain model parameters that
cause RF_aci uncertainty but are not constrained by a combination of Northern Hemisphere and pristine Southern
Ocean measurements. Secondly, even within the considerably reduced volume of multi-dimensional parameter
space there still exist many compensating parameter effects (Fig. S3), which limit the constraint on individual
parameter ranges (Lee et al., 2016; Regayre et al., 2018). The impact of these compensating effects could be
greatly reduced by perturbing uncertain emissions regionally rather than globally as we do here. Our results are
based on uncertainty in a single climate model. The model is structurally consistent in our experiments, so
neglects uncertainty caused by choice of microphysical and atmospheric process representations. Our model
also neglects some potentially important sources of remote marine aerosol, such as primary marine organic
aerosol (Mulcahy et al., 2020) and methane-sulfonic acid (Schmale et al., 2019; Hodshire, et al., 2019; Revell et
al., 2019). Model inter-comparison projects (such as CMIP6) can be used to quantify the diversity of RF (or
ERF) output from models, but they lack information about single model uncertainty. Ideally, multi-model
ensembles would contain a perturbed parameter component, so that model diversity and single model forcing
uncertainty could be quantified simultaneously. But, computational costs prevent many modelling groups from
engaging with this important aspect of uncertainty quantification, limiting our shared knowledge about the
causes of aerosol forcing uncertainty. Studies like ours that quantify the remaining uncertainty in aerosol forcing
and its components after constraint using multiple measurement types fill an important knowledge gap. This
knowledge can be used to form a more complete understanding of the importance of historical and near-term
aerosol radiative forcing which would reduce the diversity in equilibrium climate sensitivity across models.

**Data availability**

The ACE-SPACE data are accessible from: https://zenodo.org/communities/spi-ace. The basis for our cloud
droplet number concentration data are available from
http://catelogue.ceda.ac.uk/uuid/cf97ccc802d348ec8a3b6f2995dfbbff. Simulation output data for both AER and
AER-ATM PPEs are available on the JASMIN data infrastructure (http://www.jasmin.ac.uk). Some of the
climate-relevant fields are derived and stored in netCDF files (.nc) containing data for all ensemble members
and made available as a community research tool as described in Yoshioka et al. (2019). Model data and
analysis code can be made available from the corresponding author upon request.

**Author Contribution**

LR applied the statistical methodology and generated results. LR and MY created the PPEs. LR and JJ designed
the experiments and elicited probability density functions of all aerosol parameters. KC and MY participated in
the formal elicitation process. JS, AB, MG, CT, SH and FS collected and processed the ACE-SPACE
measurements. DG processed the cloud droplet number concentration data. LR, KS, JS and JJ analysed the
results. LR and KS wrote the manuscript with contributions from all authors.

**Competing Interests**

Author KC is an executive editor of ACP.

**Acknowledgements**

We acknowledge funding from NERC under grants AEROS, ACID-PRUF, GASSP and A-CURE
(NE/G006172/1, NE/I020059/1, NE/J024252/1 and NE/P013406/1) and the European Union ACTRIS-2 project
under grant 262254. This work and its contributors (LR, JJ and KC) were supported by the UK-China Research
& Innovation Partnership Fund through the Met Office Climate Science for Service Partnership (CSSP) China
as part of the Newton Fund. MY and KS received funding from the National Centre for Atmospheric Science
(NCAS), one of the UK Natural Environment Research Council (NERC) research centres via the ACSIS long-
term science programme on the Atlantic climate system. LR was funded by a Natural Environment Research
Council (NERC) Doctoral Training Grant, and a CASE studentship with the UK Met Office Hadley Centre. KC
was a Royal Society Wolfson Merit Award holder during this research. ACE-SPACE, JS, SH and AB received
funding from EPFL, the Swiss Polar Institute and Ferring Pharmaceuticals. ACE-SPACE was carried out with
additional support from the European FP7 project BACCHUS (grant agreement no. 49603445). CT was
supported by the Deutsche Forschungsgemeinschaft (DFG) in the framework of the priority programme
"Antarctic Research with comparative investigations in Arctic ice areas" SPP 1158 (grant STR 453/12-1). AB
received funding from the Swiss National Science Foundation grant No. 200021_169090. DG was funded by
the NERC-funded ACSIS programme via NCAS. This work used the ARCHER UK National Supercomputing
Service (http://www.archer.ac.uk). ARCHER project allocations n02-chem, n02-NEJ024252, n02-FREEPPE
and the Leadership Project allocation n02-CCPPE were used to perform sensitivity tests and create the
ensembles. We thank Andre Welti and Markus Hartmann for CCN measurement support provided during the
ACE-SPACE campaign.

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
