# Peer review of "The value of remote marine aerosol measurements for 1 constraining radiative forcing uncertainty 2"

_Atmospheric Chemistry and Physics, 2019_

## Referee Comment (RC1) · Anonymous Referee #1 · 17 Jan 2020

The paper is an interesting attempt at constraining radiative uncertainty with a not too common approach. The paper is generally well written and the method generally adequately described. I do think however that the some of the conclusions drawn by the authors goes beyond what is supported by the data and references, in particular with respect to sea-salt parameterisation.

It may be that this is just a matter of making the assumptions more clear.

Introduction section 2. No variations in pre-industrial aerosols –> The author has included a reference in SI (Korhonen et al (2010) that shows this is still a possibility. This may be possibly be given as an example but I agree that it is likely not important.

[Figure]

Line 79. Is this RF or is it ERF?

Line 93 N700: Is it wet or dry size. Radius or diameter?

Line 109-1111 + Figure 1d. While the constrainment of the model parameterisation narrows the range of CCN concentrations and reduces the original model bias, the constrained values has a very low variability compared to the observations. While it is understandable that the combined product should may have lower deviation than both the model and measurements alone. Is it realistic that the constrained variability is so much lower compared to observations?

Figure 2. The N700 number is much lower than than CCN_0.2 Do you have any estimates for N(total sea-salt) to show that the constrained sea-salt emissions increase is indeed the cause of CCN_0.2 and not e.g. the increase in Nss-sulphate. Or the constraining of accumulation mode dry deposition. Figure 2: Does both model and measurements use the same definition of aerosol size, i.e. the same relative humidity? If the measurements is done at e.g. 80 % relative humidity and the model results use dry radius, the N700 from the model should be lower than the measurements

Line 216: Adding the NH experiment is reducing the number of constrained model versions to 0.7 % of the total. As this likely give an even more narrow range for the constrained estimate e.g. as in figure 1d. Any comments on the validity of this heavy constrainment given that it is based on a very limited amount measurements?

SI Line 106. Any estimates for the uncertainty caused by the sampling procedure?

SI: Wind speed discrepancies. I can not see that the assumption about wind speed discrepancy being unimportant is supported at all by Korhonen et al. On the contrary the main point of Korhonen et al is that even a quite modest increase in wind-speed creates a higher CCN concentration. As the wind speed in the ensembles is said to be lower than the values in ACE-SPACE and even much lower than the climatological values the unconstrained values, the unconstrained sea-salt emissions is expected

to be lower than during the campaign and even lower compared with climatological values (potentially relevant if the "NH" added constrainment use retrieved values for AOD). Any deviations for the high wind speeds would be even more deleterious for the constrainment of sea-salt emissions.

---

## Referee Comment (RC2) · Anonymous Referee #2 · 2 Mar 2020

The authors present an interesting approach by which large PPEs of the UK Hadley Centre General Environment Model (HadGEM3) are performed in order to sample the parametric uncertainty of nearly 30 model parameters to address the following questions:

i) To what extent can measurements of aerosols in pristine (natural) environments help to constrain model simulations and thereby reduce the large uncertainty in aerosol forcing? ii) What is the relative importance of measurements in remote and polluted environments for constraining the forcing uncertainty?

Using Southern Ocean ship based measurements (CCN0.2%; CCN1%; mass concen-

trations of non-sea salt sulfate in PM10 and N700) as model constraints, the authors demonstrate a reduction in model parametric uncertainty (most notably: sea spray emissions and dry deposition velocity) that in turn reduces the aerosol forcing uncertainty from the original PPE.

The paper is generally well written, however, the methodology I find to be lacking detail in key areas. The results are generally well presented, however, given the focus of the paper is on using remote marine measurements as constraints, I would like to see a more rigorous presentation of these measurements in the main study or SI.

Finally, the conclusions presented in this study are in places not supported by the data or references. I believe that this is in part due to a lack of clarity in presentation of the methodology and assumptions therein, and in part due to limitations associated with applying pre-existing PPEs that were perhaps not specifically designed to investigate the two key questions listed above.

Due to the computationally demanding nature of running such aerosol PPEs using one set to probe a number of interesting questions associated with aerosol uncertainty is understandable, as are the potential limitations. The discussion and conclusions presented by the authors could be improved by linking to any such limitations or assumptions.

General comments:

1. Fig. 1(b/d), please overlay measured standard deviation as dots, as performed for the average of the measurements (a/c).

2. The authors are focussing on natural aerosol. How were any ship measurements influenced by anthropogenic pollution eliminated from the analysis?

3. Is each measurement used given equal weighting in constraining the model?

4. SI: The authors state: "The variance terms in the denominator of Eq. (1) are calculated uniquely for each measurement. Following Johnson et al., (2019), we use a measurement uncertainty of 10%". Are the measurement errors for the constraints used in this study homoscedastic or heteroscedastic? Do they correspond with the definition of the implausibility metric (eq. 1, SI)? How does the variability in the measurements compare to the uncertainty chosen (10%)?

5. CCN0.2% and CCN1% are used as observational constrains in the study. The measurement study in which these constraints were taken from measured CCN at more than two supersaturations. Why was a CCN spectra (or measured aerosol size distribution) not used from the observations to provide a tighter constraint on the model?

6. Please provide more detail on the observations used as constraints in the SI, linking clearly to Fig. 1 in the main article. For example, demonstrate a time-series of one of the observation dots in Fig. 1 graphically, including the variability (bars representing standard deviation), and colour of dotted time-series representing position. Clearly link this graphic to the mathematical construction of the model constraint e.g. implausibility metric in the SI.

7. The authors use four measurements as a constraint (listed above). Which measurements provided the highest information content for model constraint? I would like to see some discussion on the relative constrain the individual measurement parameters provided o the model. This would help inform future measurement campaigns in this region on key measurement parameters. For example, the authors state (SI): "Non-sea-salt sulfate was calculated by subtracting this fraction from the total particulate sulfate". How much extra constraint on the parameters (Fig. 3) is provided by using both N700 and Nss-sulfate as constraints, over just one of these.

8. The authors provide the unconstrained and constrained model PDFs of the aerosol properties. A uniform prior range is assumed in this method. How does this represent the observations? Please show a PDF of the observed distributions to see if this is a true representation of the ship observations.

9. The authors have shown how the aerosol parameters are constrained using observations, and subsequently the reduction in forcing uncertainty from the original PPEs. The paper is missing some discussion on the linkage between the constraint of these parameters and forcing. Inclusion of this would be very beneficial to the community. For example, how has average cloud microphysical properties –e.g. cloud droplet concentrations been constrained following the constraints shown in Fig. 2? Do they compare better, or worse with satellite observations in the region? This would help inform whether the constrain on forcing represents a true constraint on the aerosol processes (i.e. is the constraint of CCN by scaling sea salt right for the right reasons, or should the results be presented/interpreted as a tuning...?).

10. What is the average supersaturation over the Southern Ocean simulated by the model? How does this correspond with the selected value of CCN0.2% as representative for (cloud-active aerosol, SI) in the region?

11. The authors make clear that they are targeting parametric uncertainty, and the method does not address model structural uncertainty. However, some of the conclusions presented rely too heavily on the information provided by the parametric uncertainty analysis alone, specifically in the comparison to Revell et al., (2019) (Line 166 and thereafter). The differences in conclusions related to over/underestimation of sea spray aerosol are attributed to a lack of sampling of aerosol processes by Revell et al., 2019. A discussion on the role of structural errors in the model used by the author would be is required. What are the key differences between the model configurations with respect to representation of marine aerosol sources and sinks? What is the relevant contribution to aerosol mass from secondary vs. primary marine aerosol sources in the two model configurations?

12. Given the use of an older configuration of the model HadGEM by the authors, the results should be presented in light of the latest configuration. Stars showing the values for the parameters overlaid on Fig.3/5 that represent the configuration used by Revell et al., 2019 should be included to aid the reader in understanding differences found between the two studies with regard to sea salt emissions.

13. How much of the constraints found in Fig.3 are due to compensating parameters across the multi-dimensional marginal probability distributions? For example, what is the relationship between the marginal distributions between dry deposition and sea salt? Could the authors also provide an investigation of the joint marginal histograms between DMS and sea salt emission.

14. It is stated that the "model-measurement comparison is improved when aerosol number concentrations and mass are relatively high". Does the model configuration used have the same total sources of aerosol number/mass compared to the configuration of the model used by Revell et al., 2019? This could be included in the SI. Are there any other potential marine aerosol sources currently missing in the model configuration used by the authors that would increase aerosol number/mass by a similar magnitude than scaling sea salt emissions to 3 times the default value? This requires discussion, in particular in light of the conclusions presented by the study cited for the source of the observations (Schmale et al., 2019) used by the authors, e.g.:

Schmale et al., 2019: "The regions of highest underestimation are close to the coast of Antarctica during leg 2, close to South Africa and around 45°E during leg 1. These regions coincide with the highest concentrations of gaseous MSA . . . This preliminary model–measurement comparison suggests that the model may be missing an important source of high-latitude CCN."

15. SI: The authors state that the wind speed discrepancies do not affect the results presented. This is an important statement that deserves more detailed justification as I currently do not see how this is supported by the data or Korhonen et al., 2010. How do the differences in simulated and observed wind-speeds relate to the scaling of sea salt required to constrain CCN?

16. SI: The authors nudge the models to 2008 meteorology from reanalysis data. A comparison between the meteorological data between the measurement years and that used in the model simulation should be provided in the SI, comparing both monthly

averages and variability.

17. SI "Marginal parameter distributions are constrained consistently when we remove measurements with average wind speed differences larger than 50% of the measured value from the model-measurement comparison." How many results does this effect? Please show a global map where the grid-box colour represents a measure of how often this threshold is exceeded.

Technical comments:

1. Line 178: "The constraint on the scaled DMS emission flux is two-sided, 179 reducing the credible range of DMS emission scaling from 0.5 to 2.0 down to 0.54 to 1.9." Could the authors please make clear what in the figure 0.54/1.9 corresponds to.

2. SI, Line 95: Grammar - "pdfs with centralised tendencies will by heavily weighted". Change by to be.

3. SI, Line 63: "We make use of the ATM and AER-ATM perturbed parameter ensembles (PPEs)". Following this the authors refer only to AER and AER-ATM. Should this read: "We make use of the AER and AER-ATM"?

4. Fig. 2: Should y-axis density not be labelled 0-1? Or are these not normalised marginal densities.

---

## Author Comment (AC1) · 29 May 2020

We thank the reviewers for their thoughtful comments on our paper. We have adapted our article in response to many of the helpful comments and suggestions. In particular, we have added detail to the description of our method, including additional SI tables and figures. We have also included an additional table in the main article to show the process-based nature of our constraint on aerosol forcing. Many of our assumptions and their implications are now described in more detail, as suggested.

In the following response, we precede each reviewer comment with the abbreviation "RC:".

[Figure]

Response to anonymous reviewer 1:

RC: Line 79. Is this RF or is it ERF? We make use of 2 PPEs in this study, one which outputs aerosol RF (the AER PPE) and one which outputs aerosol ERF (AER-ATM), as described in the 3rd paragraph of the introduction (line 56 onwards).

These PPEs were designed to complement one another and are described in full in Yoshioka et al. (2019). The AER-ATM PPE used in Regayre et al. (2018) to diagnose ERF includes rapid atmospheric adjustments and perturbations to multiple physical atmosphere parameters, alongside perturbations to aerosol parameters, with horizontal wind fields nudged only above the boundary layer. However, the AER PPE used in Johnson et al. (2019) is nudged throughout the atmosphere to suppress meteorological effects entirely. Hence, no rapid adjustments were included in this PPE meaning that only aerosol RF can be analysed and constrained in this case. In this article we aim to complement and extend the constraint using output from the AER PPE from Johnson, et al. (2019) with remote marine aerosol measurements, so we mainly refer to aerosol RF. We refer to ERF when referring to corresponding results from the AER-ATM PPE.

For additional clarity, we have added "(neglecting rapid adjustments)" to the text describing the Johnson et al. (2019) constraint on around line 75 and more fully describe the efficacy of previous constraints on components of aerosol forcing: "Previous analysis of HadGEM3 PPEs showed that measurements of the present-day atmosphere in regions affected by anthropogenic emissions help to constrain the uncertainty in aerosol-radiation interaction forcing (RFari) but not the component due to aerosol-cloud interactions (RFaci). For example, Regayre et al. (2018) showed that top-of-the-atmosphere shortwave radiation flux measurements reduce ERFaci uncertainty by only around 10%, despite the fluxes in the present-day and early-industrial environments sharing multiple causes of uncertainty. Johnson et al. (2019) showed that a much larger dataset of over 9000 (predominantly Northern Hemisphere) aerosol measurements reduced the uncertainty in global, annual mean aerosol RFari (neglecting

rapid adjustments) by 35%, but RFaci uncertainty by only around 7%."

RC: Line 93 N700: Is it wet or dry size. Radius or diameter?

We use dry aerodynamic particle diameters from the Schmale et al. (2019) dataset.

We have clarified this for the reader by changing the text to "with dry aerodynamic diameter (N700; corresponds to volume equivalent diameter larger than around 500 to 570 nm; Schmale et al., 2019a)" on line 93 of the main article, line 113 of the SI and in the caption of figure 1.

RC: Line 109-1111 + Figure 1d. While the constrainment of the model parameterisation narrows the range of CCN concentrations and reduces the original model bias, the constrained values has a very low variability compared to the observations. While it is understandable that the combined product should may have lower deviation than both the model and measurements alone. Is it realistic that the constrained variability is so much lower compared to observations?

To clarify, Fig 1d is a map of the standard deviation of the monthly mean CCN concentration. It doesn't show model variability. We are aware that the use of point measurements to constrain monthly mean fields of CCN introduces temporal and spatial uncertainties, but we account for these uncertainties in our implausibility metric (through the Var(R) term in SI equation 1). Although we constrain monthly mean uncertainties in each model gridbox by more than half, the remaining uncertainties at the model gridbox scale are non-negligible and of the remaining uncertainty is the same order of magnitude as the gridbox means.

RC: Figure 2. The N700 number is much lower than than CCN_0.2 Do you have any estimates for N(total sea-salt) to show that the constrained sea-salt emissions increase is indeed the cause of CCN_0.2 and not e.g. the increase in Nss-sulphate. Or the constraining of accumulation mode dry deposition.

We think the reviewer is referring to line 162 of original manuscript, which read: "the

consistency of constraint of CCN0.2 and N700 towards higher values (Fig. 1) implies that a general scaling of the existing sea spray flux is consistent with the measurements without the need for an additional source of fine-mode, organic-rich particles.". There are many ways to combined multiple uncertain processes and get approximately the same outcome. Our figure 2 (incorrectly referred to as Fig. 1 in the original manuscript) shows that CCN0.2, N700 and nss-sulfate concentrations are all constrained to higher values. We relied on preliminary work on understanding the effects of constraining individual measurement types on model parameters to inform our analysis. However, we did not make this evidence available to the reader. We have added a table to the SI (table S3) which provides information about the effect of each measurement type constraint on model parameters. This table shows higher values of the sea spray emission flux scale factor parameter are consistent with CCN0.2 measurements.

We introduce table S3 in the SI section "SI Results: Constrained marginal parameter distributions: "In addition to the constraint achieved by combining remote marine aerosol measurements, table S3 shows the effect of individual measurement type constraints (table S2) on model parameters and how these translate into a combined constraint (Fig. 3)."

We now refer to the new table S3 in the following places.

The Fig. 3 caption, which now reads: "Fig. 3. Marginal probability distributions for the 26 aerosol parameters after constraint using ACE-SPACE measurements. The density of parameter values in the unconstrained sample are shown as horizontal dashed lines (uniform sampling over the parameter space). Densities of constrained samples are shown in colour and are scaled so that the maximum densities in the constrained and unconstrained samples are aligned. The 25th, 50th and 75th percentiles of each marginal distribution are shown in the central boxes. Parameter values on the x-axes correspond to values used in the model (Yoshioka et al., 2019, table S3)."

Around line 153 which now reads: "These joint constraints (see also Fig. S3) suggest

the model-measurement comparison is improved when aerosol number concentrations and mass are relatively high."

Around line 160 of the manuscript, which now reads: "We do not make any assumptions about the composition of these additional summertime sea spray particles. They may be rich in organic material as proposed by Gantt et al. (2011) which would alter the CCN activity of emitted particles. However, the consistency of constraint of CCN0.2 and N700 towards higher values (Fig. 2, table S3) implies that a general scaling of the existing sea spray flux is consistent with the measurements from December to April, without the need for an additional source of fine-mode, organic-rich particles."

and on line 275 which reads: "Secondly, even within the considerably reduced volume of multi-dimensional parameter space there still exist many compensating parameter effects (Fig. S3), which limit the constraint on individual parameter ranges (Lee et al., 2016; Regayre et al., 2018). The impact of these compensating effects could be greatly reduced by perturbing uncertain emissions regionally rather than globally as we do here."

RC: Figure 2: Does both model and measurements use the same definition of aerosol size, i.e. the same relative humidity? If the measurements is done at e.g. 80 % relative humidity and the model results use dry radius, the N700 from the model should be lower than the measurements

Yes. We use measured N700 concentrations of particles with dry aerodynamic diameters (40% relative humidity at the APS device air intake valve) larger than 700 nm. The volume equivalent diameter of these particles is around 500 to 570 nm. Aerosol concentrations are also calculated using dry particle diameters.

The N700 description from line 93 of the original manuscript now reads: "number concentrations of particles with dry aerodynamic diameter larger than 700 nm (N700; corresponds to volume equivalent diameter larger than around 500 to 570 nm; Schmale et al., 2019a)."

[Figure]

RC: Line 216: Adding the NH experiment is reducing the number of constrained model versions to 0.7 % of the total. As this likely give an even more narrow range for the constrained estimate e.g. as in figure 1d. Any comments on the validity of this heavy constrainment given that it is based on a very limited amount measurements?

The constrained sample contains 700 model variants, which is far more than are typically used to quantify model uncertainty or multi-model diversity. We don't agree that the measurements are "very limited". We used over 9000 measurements from Johnson et al. (2019) combined with hundreds of measurements for four aerosol properties covering much of the Southern Ocean. The results is 'valid' in the sense that these are the model variants that are most consistent with this very large set of measurements.

We have contextualised this for the reader on line 216 which now reads: "Around 700 model variants (0.07%) are observationally plausible in both the Southern Ocean (ACE-SPACE) and Johnson et al. (2019) constraints. Although this is a relatively small percentage of the original sample, 700 observationally-plausible model variants is far more than are typically used to quantify model uncertainty or multi-model diversity (e.g. around 30 for CMIP6)."

RC: SI Line 106. Any estimates for the uncertainty caused by the sampling procedure?

The sentence in question is "Fig. 1 shows the CCN0.2 mean and standard deviation from the unconstrained and constrained model variants". It's not clear to us what the referee means by "sampling procedure". Our statistical approach densely samples the model's uncertain parameter space. The constraint methodology accounts for multiple sources of uncertainty within the implausibility metric (equation S1; including using an emulator in place of the model). Therefore, the uncertainty in the posterior CCN distribution implicitly accounts for our sampling methodology.

We added a clarification to our emulation description on line 101 of the SI: "Some additional uncertainty is caused by emulating (rather than simulating) model output and this uncertainty is incorporated into our model-measurement constraint process

(SI Methods: Model-measurement comparisons), despite being much smaller than other sources of uncertainty (Johnson et al., 2019)."

RC: SI: Wind speed discrepancies. I can not see that the assumption about wind speed discrepancy being unimportant is supported at all by Korhonen et al. On the contrary the main point of Korhonen et al is that even a quite modest increase in wind-speed creates a higher CCN concentration. As the wind speed in the ensembles is said to be lower than the values in ACE-SPACE and even much lower than the climatological values the unconstrained values, the unconstrained sea-salt emissions is expected to be lower than during the campaign and even lower compared with climatological values (potentially relevant if the "NH" added constrainment use retrieved values for AOD). Any deviations for the high wind speeds would be even more deleterious for the constrainment of sea-salt emissions.

The reviewer is correct that the wording of this section was misleading. The reference to Korhonen et al. (2010) has now been removed. We referred to a subtle result in Korhonen et al. (2010) which our description did not make clear. We originally cited this article to point out that there are many factors other than sea spray which affect remote marine cloud condensation nuclei concentrations (52% of the CCN variability according to their research). Our approach compares in-situ measurements with monthly mean model data. In-situ measurements are inherently more variable because of differences in averaging period, and hence are more likely to include high wind speed events. Indeed, wind speed and N700 measurements from the ACE-SPACE campaign are only weakly correlated. Measured wind speed averages within regions defined by model gridboxes (as used in our comparison to monthly mean model output) are only weakly correlated with N700 measurements (Pearson correlation coefficient of around 0.2).

Our constraint methodology is designed to avoid relying on measurements that are in strong disagreement with the model, because these discrepancies are more likely to be caused by model structural errors. This approach also avoids the use of measurements where nudged meteorology causes large model-measurement discrepancies in

variables used for constraint.

We have altered the associated section of the SI (line 208 onwards) to clarify the importance of understanding these wind speed discrepancies and how we prevent measurements with high wind speed discrepancies from affecting our results: "SI Results: Wind Speed discrepancies Southern Ocean wind speeds during the ACE-SPACE expedition were often much lower than climatological mean values, but on average were higher than winds in our ensemble (Schmale et al., 2019). We account for the effects of inter-annual variability in the Var(R) term in equation S1. However, monthly mean differences between ERA-Interim wind speeds in the measurement year and the year used in the ensemble are less than 20% along the route taken by the ACE-SPACE campaign vessel (Fig. S4). The modest discrepancy in wind speeds may be important for constraining aerosol concentrations, because sea spray emissions in our model are strongly dependent on wind speeds (Gong, 2003). However, the measured wind speed and N700 values are only weakly correlated (Pearson correlation coefficient of around 0.2) when degraded to the resolution used for comparison with model output.

Our constraint process has in-built functionality that prevents the use of measurements with large model-measurement discrepancies. We tested the robustness of our constraint methodology to the discrepancy in wind speeds by neglecting around 50% of the measurements (those with the largest discrepancies between measured and AER-ATM PPE mean simulated winds) and repeating the constraint. The effects on marginal parameter and aerosol forcing constraints were negligible (not shown). The consistency of constraint, with and without measurements in locations with relatively large model-measurement wind speed discrepancies, suggests the constraint methodology is insensitive to wind speed discrepancies caused by daily wind speed variability and differences in meteorological years between model simulations and measurements."

Response to anonymous reviewer 2:

We have positively responded to many of reviewer 2's suggestions and think this has

improved our revised manuscript considerably. However, we think some of the changes suggested by reviewer 2 could mislead the reader, by giving a too simple representation of our constraint process. The reviewer encourages us to emphasise uncertainty in the measurements, yet our focus in this article is on uncertainties in the model-measurement constraint process. Hence, there are some suggestions we have not been able to accommodate.

RC: Fig. 1(b/d), please overlay measured standard deviation as dots, as performed for the average of the measurements (a/c).

We appreciate the motivation behind this suggestion, but think adding these dots would confuse the reader. We overlaid the mean measurements over the mean model field in Fig 1 a) and c) because these values are directly comparable. However, the model standard deviation represents uncertainty in the model parameters while the measurement standard deviation represents temporal and spatial variability, as well as instrument error – they should not be compared. We have not neglected the measurement variability. The implausibility metric used in our model-measurement constraint process includes spatial and temporal representation errors, emulation errors, inter-annual variability and instrument uncertainty. It would be misleading to compare any one of these with the model parametric uncertainty.

RC: The authors are focussing on natural aerosol. How were any ship measurements influenced by anthropogenic pollution eliminated from the analysis?

The measurements were filtered to ensure that they are free of ship stack contamination. In appendix A of Schmale et al., 2019, we explain our method: "Equivalent black carbon, trace gases data such as CO and CO2, and the 10 s-1 variability of particle number concentrations were used to identify the influence of ship exhaust. Identified exhaust periods are not included here and constitute about 50% of the total data. Size-dependent particle losses in the inlet lines were determined experimentally after the cruise and data are corrected accordingly. Losses were <10% for submicron particles

and about 15% for supermicron particles.".

It is important to note that we don't assume that all the sampled aerosol was natural. The atmosphere may have contained some anthropogenic aerosol from distant sources. The model includes these and several anthropogenic aerosol emission and process parameters were perturbed in our ensemble. These parameters were very modestly constrained, suggesting that the environment is dominated by natural aerosol.

RC: Is each measurement used given equal weighting in constraining the model?

Yes, measurement types are given as equal a weighting as possible in the constraint process.

The description of our constraint process in the original manuscript lacked some important details and relied on the methodology described in Johnson et al. (2019). Our constraint process relies on the use of "implausibility" metrics, which are calculated for each of the one million model variants, for each measurement type at each measurement location. We set implausibility thresholds for each measurement type and also set exceedance tolerances, defined as the number (or percentage) of measurements for which a model variant's output exceeds the specified threshold. The constraint efficacy differs between measurement types and we adjust the tolerance and exceedance threshold values (defined in the SI) for each measurement type to ensure each variable constraint retains approximately the same proportion of the original sample of model variants. The proportion retained by individual measurement type constraints varies from 18% to 30%.

We have enhanced the description of our implausibility threshold and exceedance tolerance value selection process in the SI section "SI Methods: Model-measurement comparisons" and have included two additional tables (tables S1 and S2) for readers interested in the specific values used for each measurement type.

The additional text included in the adjusted SI reads: "We set threshold and tolerance values for each variable distinctly for each month of data. This makes processing the implausibility data more efficient and allows for a degree of automation of the constraint process. We ensure that each measurement type on each leg of the journey (Schmale et al., 2019) affects the combined constraint. This requires quantification of the constraint of individual measurement types on parameter values at multiple combinations of threshold and implausibility exceedance tolerances. We avoid increasing the threshold and/or tolerance values in individual months for each measurement type, if the constraint efficacy of the measurement would saturate as a result. Otherwise, threshold and tolerances for each month are required to be as similar as possible.

Although our analysis in the main article focusses on a combined measurement constraint, this analysis is informed by individual measurement type constraints. The threshold and exceedance tolerances for individual measurement type constraints are summarised in table S1. Only 0.004% of the one million model variants (40 variants) are retained when these individual constraints are combined. Thus, we relax the threshold and tolerance criteria for each measurement type constraint when combining constraints (table S2)."

RC: SI: The authors state: "The variance terms in the denominator of Eq. (1) are calculated uniquely for each measurement. Following Johnson et al., (2019), we use a measurement uncertainty of 10%". Are the measurement errors for the constraints used in this study homoscedastic or heteroscedastic? Do they correspond with the definition of the implausibility metric (eq. 1, SI)? How does the variability in the measurements compare to the uncertainty chosen (10%)?

We applied heteroscedastic uncertainties for measurement and representation errors for consistency with Johnson et al. (2019). It would have been far simpler to apply homoscedastic uncertainties. We acknowledge that our choice of heteroscedastic errors is a subjective decision. However, as shown in Fig. 2, we reject model variants with the lowest values for each measurement type, which correspond to our lowest

instrument error values. If we had used homoscedastic errors, all constraints would have been weaker. Thus, we would have needed to reduce implausibility thresholds and exceedance tolerances to attain the same degree of constraint.

We think the 10% instrument error applied here is an overestimate. This is a cautious approach that allows us to avoid over-constraint based on this set of measurements. Furthermore, there is limited data available to inform our choice of spatial and temporal representation errors. The "variability in the measurements" on short timescales at point locations conflates instrument error with spatial and temporal representation errors, but does not fully encompass any of these. Dedicated measurement campaigns are required to establish robust estimates of these errors. We therefore elected to use 10% of the measured value for instrument error, as well as 20% and 10% respectively for spatial and temporal representation errors, to maintain consistency with Johnson et al. (2019). These errors are typically larger than strictly necessary, which is intentional. Larger uncertainties prevent us from over-constraining the model. Our approach is based on ruling out model variants (and parts of parameter space) that are implausible, rather than on finding all model variants that are plausible. This is a subtle, but important, distinction that shapes our methodology. Even so, using these relatively large model-measurement comparison uncertainties, we are able to rule out the majority of model variants successfully by adjusting our implausibility thresholds and exceedance tolerances.

RC: CCN0.2% and CCN1% are used as observational constrains in the study. The measurement study in which these constraints were taken from measured CCN at more than two supersaturations. Why was a CCN spectra (or measured aerosol size distribution) not used from the observations to provide a tighter constraint on the model?

It is reasonable to assume we would attain a stronger constraint if we had used the full CCN spectra. However, we showed in Johnson et al. (2018) that diverse measurements are more useful for constraint than additional measurements of a similar nature.

Hence, we elected to combine measured concentrations of N700 and non-sea-salt sulfate with CCN concentrations at two supersaturatons that provide distinct information about the aerosol size distribution, rather than multiple supersaturations that would provide similar information and constraints.

We actually found that the CCN0.2 and CCN1.0 measurements provide very similar constraints on the model parameters, so we do not have cause to believe additional CCN measurements at alternative supersaturations would improve the constraint. We have included an additional table (table S3) in the SI showing how individual measurement types constrain the parameters, so that the reader can appreciate the similarity of CCN constraints at different supersaturations.

RC: Please provide more detail on the observations used as constraints in the SI, linking clearly to Fig. 1 in the main article. For example, demonstrate a time-series of one of the observation dots in Fig. 1 graphically, including the variability (bars representing standard deviation), and colour of dotted time-series representing position. Clearly link this graphic to the mathematical construction of the model constraint e.g. implausibility metric in the SI.

The measurements used in our constraint are publically available (https://zenodo.org/communities/spi-ace?page=1&size=20%20) and we reference the link to the dataset in Schmale et al. (2019), as well as here in the appendix on line 293.

It is important to note that our implausibility metric relies on multiple sources of model-measurement comparison uncertainty and the reviewer's request highlights only one (limited) aspect of the uncertainty. However, we think showing how the high time resolutions data is degraded for comparison to model output warrants attention. Therefore, we have included an additional figure in the SI (Fig. S1) which gives an example of this process. We added this figure to the section "SI Methods: Model-measurement comparisons" on around line 132, so that the relevance of degrading measurements

for comparison with model output using our implausibility metric is contextualised.

The revised SI text referring to Fig. S1 reads: "The variance terms in the denominator of Eq. (1) are calculated uniquely for each measurement. Following Johnson et al. (2019), we use an instrument error of 10%, a spatial co-location uncertainty of 20% and a temporal co-location uncertainty of 10%. Fig. S1 shows an example of the degradation of data for comparison with monthly mean model output. Emulator uncertainty is calculated for each model-measurement combination using the error on the predicted mean from the emulator for the model variant. We use residuals in de-trended monthly mean output from a HadGEM-UKCA hindcast simulation over the period of 1980-2009 (Turnock et al., 2015) to estimate the inter-annual variability for each variable across all model gridboxes and months."

RC: The authors use four measurements as a constraint (listed above). Which measurements provided the highest information content for model constraint? I would like to see some discussion on the relative constrain the individual measurement parameters provided o the model. This would help inform future measurement campaigns in this region on key measurement parameters. For example, the authors state (SI): "Non-sea-salt sulfate was calculated by subtracting this fraction from the total particulate sulfate". How much extra constraint on the parameters (Fig. 3) is provided by using both N700 and Nss-sulfate as constraints, over just one of these

We agree with the reviewer. The efficacy of individual measurement type constraints on model parameters and processes is important and could be used to motivate targeted measurement campaigns. This information could also help identify model development priorities. We have therefore included an additional table in the SI (table S3) that shows the effects of individual measurement type constraints on model parameters. Significant additional effort would be needed to quantify the 2-way and 3-way constraint combinations. Our research focus in this paper is on the benefits of the ACE-SPACE measurements over and above more readily available measurements, so we have only added the requested individual constraints, which we think will satisfy the curiosity of

the vast majority of readers.

In addition to table S3, we have added the following explanatory text in the SI: "In addition to the constraint achieved by combining remote marine aerosol measurements, table S3 shows the effect of individual measurement type constraints (table S2) on model parameters and how these translate into a combined constraint (Fig. 3)."

RC: The authors provide the unconstrained and constrained model PDFs of the aerosol properties. A uniform prior range is assumed in this method. How does this represent the observations? Please show a PDF of the observed distributions to see if this is a true representation of the ship observations.

We think there has been some confusion. The uniform prior ranges are applied to individual uncertain model parameters (not variables) and are used to densely sample model uncertainty (one million model variants) uniformly across the multi-dimensional parameter space using our statistical emulators. This unconstrained sample results in pdfs of the output variables shown in Fig. 2. We make no assumptions about measurement distributions, except for the model-measurement comparison uncertainties included in equation S1.

RC: The authors have shown how the aerosol parameters are constrained using observations, and subsequently the reduction in forcing uncertainty from the original PPEs. The paper is missing some discussion on the linkage between the constraint of these parameters and forcing. Inclusion of this would be very beneficial to the community. For example, how has average cloud microphysical properties –e.g. cloud droplet concentrations been constrained following the constraints shown in Fig. 2? Do they compare better, or worse with satellite observations in the region? This would help inform whether the constrain on forcing represents a true constraint on the aerosol processes (i.e. is the constraint of CCN by scaling sea salt right for the right reasons, or should the results be presented/interpreted as a tuning...?).

We agree that this is an important consideration, which will help the reader understand

that our method leads to an actual constraint on model output, which a typical model tuning approach would not. Our method relies on ruling out implausible model variants (a true constraint), rather than identifying the best model (a tuning process). Therefore, processes are constrained as the reviewer suggests. We have described these results and added a new table to the main article (table 1). We also include an additional co-author (Daniel P. Grosvenor) who provided cloud droplet number concentration data for analysis. The text at around line 191 now reads:

"Firstly, the magnitude of median RFaci weakens from -1.99 W m-2 to -1.88 W m-2 (-1.64 to -1.49 W m-2 for ERFaci). A weaker forcing is consistent with higher natural aerosol emissions, increased aerosol load and higher cloud droplet number concentrations in the early-industrial period. Table 1 shows that our constraint on natural emission parameters also constrains Southern Ocean cloud droplet number concentrations towards higher values, reducing the credible interval by around 50% and bringing mean values into closer agreement with MODerate Imaging Spectroradiometer (MODIS; Salomonson et al., 1989) instrument data (note that droplet number concentrations were not used to constrain the model). Thus, we conclude that the constraint on aerosol forcing towards weaker values is a genuine constraint and not the result of an arbitrary tuning."

We have also described how we processed the cloud droplet number concentrations in the SI section "SI Methods: Measurements" on line 108 of the SI: "We present monthly mean and annual cloud droplet number concentrations in table 1 from the model and from satellite data, over the region between 50oS and 60oS. Following Grosvenor et al., (2018), we calculated cloud droplet concentrations from the MODIS (MODerate Imaging Spectroradiometer) Collection 5.1 Joint Level-2 (Aqua satellite) for the year 2008 (to correspond to the meteorological year used in our simulations). Our calculation used cloud optical depth and 3.7 micron effective radius values derived under the adiabatic cloud assumption (essentially, cloud liquid water increases linearly with height, droplet concentrations are constant throughout the cloud and the ratio of

volume mean radius to effective radius is constant). We improved the cloud droplet
concentration data (Grosvenor et al., 2018b) by excluding 1x1 degree data points for
which the maximum sea-ice areal coverage over a moving 2-week window exceeded
0.001%. The sea-ice data used in this process were the daily 1x1 degree version of
Cavalieri et al. (2016). As with other data used in our model-measurement comparison,
we degraded the cloud droplet number concentration data to the model gridbox and
monthly mean spatial and temporal resolutions."

Finally, we state how the cloud droplet number concentration data
can be accessed in the "Data availability" section: "The basis for
our cloud droplet number concentration data are available from
http://catelogue.ceda.ac.uk/uuid/cf97ccc802d348ec8a3b6f2995dfbbff."

RC: What is the average supersaturation over the Southern Ocean simulated by the
model? How does this correspond with the selected value of CCN0.2% as representa-
tive for (cloud-active aerosol, SI) in the region?

Cloud supersaturation is not known from measurements. The measurements of CCN
at 0.2 and 1.0% supersaturation span a range of likely values. The key point here is
that both CCN definitions constrain the model quite similarly (now made clear to the
reader in table S3), so it is not vital that we know the actual supersaturation precisely.

RC: The authors make clear that they are targeting parametric uncertainty, and the
method does not address model structural uncertainty. However, some of the conclu-
sions presented rely too heavily on the information provided by the parametric uncer-
tainty analysis alone, specifically in the comparison to Revell et al., (2019) (Line 166
and thereafter). The differences in conclusions related to over/underestimation of sea
spray aerosol are attributed to a lack of sampling of aerosol processes by Revell et al.,
2019. A discussion on the role of structural errors in the model used by the author
would be is required. What are the key differences between the model configurations
with respect to representation of marine aerosol sources and sinks? What is the rele-

vant contribution to aerosol mass from secondary vs. primary marine aerosol sources in the two model configurations?

We agree with the reviewer. Many readers will be interested in how repeating our analysis using a model that includes structural developments might affect our results. We encourage that sort of activity. Therefore, we have added detail to our description (in the discussion section) of the importance of interpreting our results in the context of single climate model uncertainty. We have also made it clear that our method neglects structural uncertainties. This adds to our discussion about the need to quantify both single model uncertainty and multi-model diversity in our conclusions. We have not contrasted the primary vs secondary contributions to marine aerosols in the models because this goes well beyond the scope of our article. However, we now highlight some of the structural advances most likely to affect our results in the paragraph starting on line 281 of the original manuscript.

The revised text reads: "Our results are based on uncertainty in a single climate model. The model is structurally consistent in our experiments, so neglects uncertainty caused by choice of microphysical and atmospheric process representations. Our model also neglects some potentially important sources of remote marine aerosol, such as primary marine organic aerosol (Mulcahy et al., 2020) and methane-sulfonic acid (Schmale et al., 2019; Hodshire, et al., 2019; Revell et al., 2019). Model inter-comparison projects (such as CMIP6) can be used to quantify the diversity of RF (or ERF) output from models, but they lack information about single model uncertainty. Ideally, multi-model ensembles would contain a perturbed parameter component, so that model diversity and single model forcing uncertainty could be quantified simultaneously. But, computational costs prevent many modelling groups from engaging with this important aspect of uncertainty quantification, limiting our shared knowledge about the causes of aerosol forcing uncertainty. Studies like ours that quantify the remaining uncertainty in aerosol forcing and its components after constraint using multiple measurement types fill an important knowledge gap. This knowledge can be used to form a more complete

understanding of the importance of historical and near-term aerosol radiative forcing which would reduce the diversity in equilibrium climate sensitivity across models. "

It has been brought to our attention that we misinterpreted the results in Revell et al. (2019), by misreading the seasonal effects described in the article. Revell et al. (2019) showed, using a more recent version of our model and using interactive chemistry, that simulated sea spray aerosol concentrations are higher than observed in Jun-Aug when wind speeds are relatively high. However, in Dec-Feb the model simulates too-low cloud droplet number concentrations and AOD compared with satellite observations. Our article text has been adapted to more accurately represent the consistency of our constraint with the findings of Revell et al. (2019), and to more transparently describe the seasonal specificity of our constraint on sea salt emissions using the Gong (2003) parametrisation.

The adapted text on line 158 reads: "This suggests that sea spray emissions in our model need to be significantly higher than those calculated using the wind speed dependent Gong (2003) parametrisation in the Southern Hemisphere summer. The higher flux is consistent with Revell et al. (2019), who showed that a more recent version of our model simulates cloud droplet concentrations and aerosol optical depth values that are lower than observed over the Southern Ocean in the Southern Hemisphere summer. However, in the Southern Hemisphere winter Revell et al., (2019) simulated higher aerosol optical depths than observed, which they corrected by reducing the dependence of sea spray emissions on wind speed. Hence, our constraint on sea spray emission fluxes may only be appropriate for Southern Hemisphere summer when wind speeds are relatively low. We do not make any assumptions about the composition of these additional summertime sea spray particles. They may be rich in organic material as proposed by Gantt et al. (2011) which would alter the CCN activity of emitted particles. However, the consistency of constraint of CCN0.2 and N700 towards higher values (Fig. 2, table S3) implies that a general scaling of the existing sea spray flux is consistent with the measurements from December to April, without

the need for an additional source of fine-mode, organic-rich particles."

RC: Given the use of an older configuration of the model HadGEM by the authors, the results should be presented in light of the latest configuration. Stars showing the values for the parameters overlaid on Fig.3/5 that represent the configuration used by Revell et al., 2019 should be included to aid the reader in understanding differences found between the two studies with regard to sea salt emissions.

This suggested change is no longer pertinent, since our results are in better agreement with Revell et al., (2019) than we initially thought. We have not included the suggested alteration to our figures, because Revell et al. (2019) made structural changes to process representations, including to the Gong (2003) sea salt emission parameterisation and model chemistry. Thus, we believe highlighting parameter values used in a structurally different model would mislead the reader. The effect of including structural changes on model output is described in Revell et al. (2019) and in the model development papers cited within.

RC: How much of the constraints found in Fig.3 are due to compensating parameters across the multi-dimensional marginal probability distributions? For example, what is the relationship between the marginal distributions between dry deposition and sea salt? Could the authors also provide an investigation of the joint marginal histograms between DMS and sea salt emission.

We agree that the joint constraint of key parameters may be of considerable interest to readers and thank the reviewer for the suggestion. In Fig. 13 of Regayre et al. (2018) we used 2-dimensional density plots to highlight the important role model equifinality plays on reducing constraint efficacy when single-model uncertainty is densely sampled. Here, we added figure S3 to the SI to show joint marginal densities of key parameters as suggested.

We introduce the new figure (Fig. S3) in the SI section "SI Results: Constrained marginal parameter distributions": "Constrained marginal parameter distributions in

Fig. 3 and Fig. 5 of the main article tell a one-dimensional story. In Fig. S3, we show the effect of constraint to remote marine aerosol measurements, combined with the constraint from Johnson et al. (2019) on a subset of the marginal 2-dimensional parameter combinations."

We refer to this new figure on around line 153 of our revised article to highlight the consistency of constraint across the parameter space and on line 275 to emphasise how compensating parameter effects limit the efficacy of constraint.

"These joint constraints (see also Fig. S3) suggest the model-measurement comparison is improved when aerosol number concentrations and mass are relatively high."

"Secondly, even within the considerably reduced volume of multi-dimensional parameter space there still exist many compensating parameter effects (Fig. S3), which limit the constraint on individual parameter ranges (Lee et al., 2016; Regayre et al., 2018)."

RC: It is stated that the "model-measurement comparison is improved when aerosol number concentrations and mass are relatively high". Does the model configuration used have the same total sources of aerosol number/mass compared to the configuration of the model used by Revell et al., 2019? This could be included in the SI.

We agree that it is important to make the reader aware of potentially important structural advances that may affect interpretation of our results, which we now do in the final paragraph of the discussion. However, we do not contrast our total aerosol and mass with those in Revell et al. (2019) for two main reasons. Firstly, our results are more consistent with those of Revell et al. (2019) than originally thought. Secondly, our article focuses on the single-model uncertainty constraint. Comparisons between model versions with structurally distinct process representations is beyond the scope of our article. This sort of analysis, based on multiple structural changes, is best presented using experiments designed for that specific purpose.

The revised text reads: "Our results are based on uncertainty in a single climate model.

The model is structurally consistent in our experiments, so neglects uncertainty caused by choice of microphysical and atmospheric process representations. Our model also neglects some potentially important sources of remote marine aerosol, such as primary marine organic aerosol (Mulcahy et al., 2020) and methane-sulfonic acid (Schmale et al., 2019; Hodshire, et al., 2019; Revell et al., 2019). Model inter-comparison projects (such as CMIP6) can be used to quantify the diversity of RF (or ERF) output from models, but they lack information about single model uncertainty. Ideally, multi-model ensembles would contain a perturbed parameter component, so that model diversity and single model forcing uncertainty could be quantified simultaneously. But, computational costs prevent many modelling groups from engaging with this important aspect of uncertainty quantification, limiting our shared knowledge about the causes of aerosol forcing uncertainty. Studies like ours that quantify the remaining uncertainty in aerosol forcing and its components after constraint using multiple measurement types fill an important knowledge gap. This knowledge can be used to form a more complete understanding of the importance of historical and near-term aerosol radiative forcing which would reduce the diversity in equilibrium climate sensitivity across models."

RC: Are there any other potential marine aerosol sources currently missing in the model configuration used by the authors that would increase aerosol number/mass by a similar magnitude than scaling sea salt emissions to 3 times the default value? This requires discussion, in particular in light of the conclusions presented by the study cited for the source of the observations (Schmale et al., 2019) used by the authors, e.g.: Schmale et al., 2019: "The regions of highest underestimation are close to the coast of Antarctica during leg 2, close to South Africa and around 45_E during leg 1. These regions coincide with the highest concentrations of gaseous MSA... This preliminary model–measurement comparison suggests that the model may be missing an important source of high-latitude CCN."

We have highlighted the potential role of marine organic material, but also stated that consistency of constraint of CCN and N700 does not suggest the need for a special

source into the accumulation mode. In Schmale et al. (2019) we drew attention to high MSA concentration measurements near the Antarctic coast. DMS emissions themselves were constrained near their central value, so do not appear to be the cause of the biases. We are not aware of any other potential explanations for such large and consistent biases in CCN and N700.

The additional text on line 281, which refers to additional sources of aerosol neglected by our experiments is: "The model is structurally consistent in our experiments, so neglects uncertainty caused by choice of microphysical and atmospheric process representations. Our model also neglects some potentially important sources of remote marine aerosol, such as primary marine organic aerosol (Mulcahy et al., 2020) and methane-sulfonic acid (Schmale et al., 2019; Hodshire, et al., 2019; Revell et al., 2019)."

RC: SI: The authors state that the wind speed discrepancies do not affect the results presented. This is an important statement that deserves more detailed justification as I currently do not see how this is supported by the data or Korhonen et al., 2010. How do the differences in simulated and observed wind-speeds relate to the scaling of sea salt required to constrain CCN?

Our reference to Korhonen et al. (2010) has now been removed. We agree that wind speeds are important for calculating sea spray emissions and did not intend to mislead the reader. We originally cited Korhonen et al. (2010) to point out that there are many factors other than sea spray which affect remote marine cloud condensation nuclei concentrations (52% of the CCN variability according to their research). Wind speeds measured during the ACESPACE campaign are only weakly correlated with measured N700 concentrations. The Pearson correlation coefficient is only 0.2 when degraded to the model gridbox scale used for comparison to model output. Also, our constraint method is designed to avoid the use of measurements where structural model errors are the cause of model-measurement discrepancies. Thus, when we pre-filter the data by removing all measurements where nudged and measured wind speeds differ

meaningfully, the constraint on parameters is unaffected. We have adapted this section of the SI to give the reader a better appreciation of why wind speed discrepancies do not affect the constraint.

The revised section "SI Results: Wind Discrepancies" reads: "Southern Ocean wind speeds during the ACE-SPACE expedition were often much lower than climatological mean values, but on average were higher than winds in our ensemble (Schmale et al., 2019). We account for the effects of inter-annual variability in the Var(R) term in equation S1. However, monthly mean differences between ERA-Interim wind speeds in the measurement year and the year used in the ensemble are less than 20% along the route taken by the ACE-SPACE campaign vessel (Fig. S4). The modest discrepancy in wind speeds may be important for constraining aerosol concentrations, because sea spray emissions in our model are strongly dependent on wind speeds (Gong, 2003). However, the measured wind speed and N700 values are only weakly correlated (Pearson correlation coefficient of around 0.2) when degraded to the resolution used for comparison with model output.

Our constraint process has in-built functionality that prevents the use of measurements with large model-measurement discrepancies. We tested the robustness of our constraint methodology to the discrepancy in wind speeds by neglecting around 50% of the measurements (those with the largest discrepancies between measured and AER-ATM PPE mean simulated winds) and repeating the constraint. The effects on marginal parameter and aerosol forcing constraints were negligible (not shown). The consistency of constraint, with and without measurements in locations with relatively large model-measurement wind speed discrepancies, suggests the constraint methodology is insensitive to wind speed discrepancies caused by daily wind speed variability and differences in meteorological years between model simulations and measurements."

RC: SI: The authors nudge the models to 2008 meteorology from reanalysis data. A comparison between the meteorological data between the measurement years and that used in the model simulation should be provided in the SI, comparing both monthly

averages and variability.

We have included an additional figure (Fig. S4) to highlight the importance of the discrepancy in meteorology. We relate the difference in meteorology to the inter-annual variability uncertainty term included in our implausibility calculations. Monthly mean data are compared, since this is the reference scale used in our model-measurement comparison. The effects of differences in daily wind speed variability between measurement year and model simulation year are included in the inter-annual variability and temporal and spatial error terms in our implausibility metric. However, we have included an additional figure S1 to exemplify the effect of degrading measurement data to the model gridbox resolution for model-measurement comparison.

Fig. S4 is referenced in the revised section "SI Results: Wind Discrepancies" which now reads: "Southern Ocean wind speeds during the ACE-SPACE expedition were often much lower than climatological mean values, but on average were higher than winds in our ensemble (Schmale et al., 2019). We account for the effects of inter-annual variability in the Var(R) term in equation S1. However, monthly mean differences between ERA-Interim wind speeds in the measurement year and the year used in the ensemble are less than 20% along the route taken by the ACE-SPACE campaign vessel (Fig. S4). The modest discrepancy in wind speeds may be important for constraining aerosol concentrations, because sea spray emissions in our model are strongly dependent on wind speeds (Gong, 2003). However, the measured wind speed and N700 values are only weakly correlated (Pearson correlation coefficient of around 0.2) when degraded to the resolution used for comparison with model output."

RC: SI "Marginal parameter distributions are constrained consistently when we remove measurements with average wind speed differences larger than 50% of the measured value from the model-measurement comparison." How many results does this effect? Please show a global map where the grid-box colour represents a measure of how often this threshold is exceeded.

We have added the requested detail to the SI text. However, at the resolution used for model-measurement comparison, the correlation between measured wind speed and N700 is near-zero (Pearson correlation coefficient of 0.2). Furthermore, our constraint methodology is insensitive to large model-measurement discrepancies caused by model structural errors. This feature was our motivation for including an SI section on wind speed discrepancies, but was inadequately described in our original manuscript. Thus, we have not included the suggested figure, which could confuse the reader by leading them to assume the N700 measurements in these locations are less reliable than they are. Instead, we have refined the "SI Results: Wind Discrepancies" text:

"Southern Ocean wind speeds during the ACE-SPACE expedition were often much lower than climatological mean values, but on average were higher than winds in our ensemble (Schmale et al., 2019). We account for the effects of inter-annual variability in the Var(R) term in equation S1. However, monthly mean differences between ERA-Interim wind speeds in the measurement year and the year used in the ensemble are less than 20% along the route taken by the ACE-SPACE campaign vessel (Fig. S4). The modest discrepancy in wind speeds may be important for constraining aerosol concentrations, because sea spray emissions in our model are strongly dependent on wind speeds (Gong, 2003). However, the measured wind speed and N700 values are only weakly correlated (Pearson correlation coefficient of around 0.2) when degraded to the resolution used for comparison with model output.

Our constraint process has in-built functionality that prevents the use of measurements with large model-measurement discrepancies. We tested the robustness of our constraint methodology to the discrepancy in wind speeds by neglecting around 50% of the measurements (those with the largest discrepancies between measured and AER-ATM PPE mean simulated winds) and repeating the constraint. The effects on marginal parameter and aerosol forcing constraints were negligible (not shown). The consistency of constraint, with and without measurements in locations with relatively large

model-measurement wind speed discrepancies, suggests the constraint methodology is insensitive to wind speed discrepancies caused by daily wind speed variability and differences in meteorological years between model simulations and measurements."

RC: Line 178: "The constraint on the scaled DMS emission flux is two-sided, 179 reducing the credible range of DMS emission scaling from 0.5 to 2.0 down to 0.54 to 1.9." Could the authors please make clear what in the figure 0.54/1.9 corresponds to.

We have now defined DMS as dimethylsulfide on line 152: "Several other parameters (related to cloud droplet pH, dimethylsulfide (DMS) emissions and wet deposition) are more modestly constrained."

We have also clarified that DMS is an aerosol precursor, and that this parameter is a scale factor on the default emissions (originally referred to as a scaling). We have now included references to the schemes used, since other modelling groups may use different seawater concentrations and/or emission flux representations.

Revised text on line 178 now reads: "Other parameters are more modestly constrained. The constraint on the aerosol precursor DMS emission flux scale factor is two-sided, reducing the credible range of DMS emission scalings from 0.5 to 2.0 down to 0.54 to 1.9. This constraint suggests the default surface sea water concentration (Kettle and Andreae, 2000) and emission parameterisation (Nightingale, et al., 2000) are consistent with measurements (including aerosol sulfate) and do not benefit from being scaled. Furthermore, ACE-SPACE measurements are consistent with less-efficient aerosol scavenging (55% likelihood of Rain_Frac, the parameter that controls the fractional area of the cloudy part of model grid boxes where rain occurs, being below the unconstrained median value 0.5) and less aqueous phase sulfate production (pH of cloud droplets has a 62% likelihood of being lower than the unconstrained median value). These combined constraints suggest, in agreement with sea spray and deposition parameter constraints, higher aerosol number and mass concentrations are consistent with measurements."

RC: SI, Line 95: Grammar - "pdfs with centralised tendencies will by heavily weighted". Change by to be.

Done.

RC: SI, Line 63: "We make use of the ATM and AER-ATM perturbed parameter ensembles (PPEs)". Following this the authors refer only to AER and AER-ATM. Should this read: "We make use of the AER and AER-ATM"?

Yes, this has been corrected.

RC: Fig. 2: Should y-axis density not be labelled 0-1? Or are these not normalised marginal densities.

Our description of these figures was inadequate. The purpose of this figure is to contrast the shape of the probability densities for the unconstrained and constrained sets of model variants. These are not normalised marginal densities. The density curve for each sample of model variants (unconstrained and constrained) is scaled such that the area under the curve integrates to one. This means that the densities can be compared visually on the same figure. The values on the y-axis are not helpful (or needed) for comparing the shape of probability density curves of the different samples, and have therefore not been included in the figure.

We have added an extra sentence to the caption of figure 2 to make the scaling clear.

"Densities for each sample of model variants are scaled so that the area under the curve integrates to one."

---

## Author Comment (AC2) · 29 May 2020

These figures and tables have all been included in response to reviewer comments. They need to be viewed alongside our responses, so that reviewers can appreciate how their comments have shaped our manuscript. Figure and table captions are included here, followed by the figures and tables themselves.

Fig S1: Measured CCN0.2 values between the 3rd and 10th January 2017, after filtering for possible ship stack contamination. The ACE-SPACE vessel transited through 5 model gridboxes during this period. We average all measurements collected in locations, over one or more days, within each model gridbox, for comparison with monthly

[Figure]

mean model output. These average values and one standard deviation of the measurement data are shown in red at the central time for each measurement subset. From left to right, these values correspond to the five model gridboxes in Fig. 1 between around 60oE and 90oE, at the following latitude and longitudes: 1) 49.5oS, 65.5oE, 2) 49.5oS, 69.5oE, 3) 54oS, 77oE, 4) 54oS, 84.5oE and 5) 56.5oS, 92oE.

Fig. S3. Two-dimensional marginal probability density distributions for a) sea spray emission flux scale factor (Sea_Spray) and the Accumulation aerosol mode dry deposition velocity scale factor (Dry_Dep_Acc), b) sea spray emission flux scale factor and dimethylsulfide surface water concentration scale factor (DMS), c) sea spray emission flux scale factor and cloud droplet pH (Cloud_pH), and d) Accumulation aerosol mode dry deposition velocity scale factor and dimethylsulfide surface water concentration scale factor. Individual parameter ranges are plotted according to their constrained values (table S3), not the full range of values used in the original sample of model variants as shown in Fig. 3, Fig. 5 and Fig. S2.

Fig. S4. Ratio of ERA-Interim wind speed differences (between measurement and simulated years) to the measurement year. Monthly mean winds from 2006 (matching the AER PPE) were subtracted from monthly mean winds for December 2016 to April 2017 (matching the ACE-SPACE campaign) to calculate the differences. The map is an assimilation of data between months, where data is presented at each location for months corresponding to the timing of the ACE-SPACE measurement campaign.

Table 1. Annual and monthly mean cloud drop number concentrations over the Southern Ocean (over the region between 50oS and 60oS at around 1km altitude above sea level) in the original unconstrained sample and the sample of model variants constrained to ACESPACE campaign measurements. Mean values and 95% credible interval values are shown for each sample, with interquartile ranges in brackets. For comparison, we show cloud drop concentrations calculated from MODIS instrument data following Grosvenor et al., (2018) for the year 2008 (SI Methods: Measurements).

Table S1: Individual measurement type constraint threshold values and exceedance tolerance values for December to April, as well as the percentage of the one million member sample retained by each constraint. Exceedance tolerances values are percentages of the number of measurements in each month.

Table S2: Threshold values and exceedance tolerance values for December to April, as well as the percentage of the one million member sample retained by each constraint. Exceedance tolerances values are percentages of the number of measurements in each month. These constraints are combined to retain around 3% of the one million member sample of model variants, as described in the main article.

Table S3. Ranges and inter-quartile ranges of marginal parameter distributions from individual constraints using measured concentrations of CCN0.2, CCN1.0, non-sea-salt sulfate and N700, as well as for the combined constraint. These individual constraints are those described in table S2 and were combined to constrain the model and make Fig. 3. Values are marked in bold where the individual measurement type constraint moves the range, 25th or 75th percentile closer towards the range or percentiles of the combined constraint than other measurement types, relative to the unconstrained values.

[Figure]

**Fig. 1.** Fig. S1. A new figure in response to reviewer comments

**Fig. 2.** Fig. S3. A new figure in response to reviewer comments

![Polar projection map of Antarctica showing ratio values with green and pink shading, circular markers, and coordinate gridlines. Color scale below ranges from -0.8 to 0.8 labeled "Ratio".]

**Fig. 3.** Fig. S4. A new figure in response to reviewer comments

| | Annual | December | January | February | March | April |
|---|---|---|---|---|---|---|
| MODIS (cm$^{-3}$) | 73 | 89 | 91 | 90 | 82 | 63 |
| Unconstrained mean (cm$^{-3}$) | 38 | 39 | 39 | 41 | 42 | 39 |
| Unconstrained credible interval (cm$^{-3}$) | 7-125 (112) | 8-115 (103) | 8-117 (109) | 7-122 (115) | 7-129 (122) | 7-118 (111) |
| Constrained mean (cm$^{-3}$) | 66 | 67 | 69 | 72 | 76 | 70 |
| Constrained credible interval (cm$^{-3}$) | 41-96 (55) | 43-96 (53) | 44-99 (55) | 45-105 (60) | 47-111 (64) | 44-101 (57) |

**Fig. 4.** Table 1. A new table in response to reviewer comments

| | CCN$_{0.2}$ | CCN$_{1.0}$ | Nss-sulfate | N$_{700}$ |
|---|---|---|---|---|
| Implausibility Threshold | 3.5 | 3.5 | 3.5 | 3.5 |
| Exceedance tolerance (%) Dec-Apr | 15,15,20,20,10 | 2,2,2,5,2 | 15,20,20,15 | 20,20,25,20,20 |
| Percentage retained | 3.3 | 3.0 | 6.2 | 3.0 |

**Fig. 5.** Table S1. A new table in response to reviewer comments

|  | CCN$_{0.2}$ | CCN$_{1.0}$ | Nss-sulfate | N$_{700}$ |
|---|---|---|---|---|
| Implausibility Threshold | 4.5 | 4.5 | 4.0 | 4.5 |
| Exceedance tolerance (%) Dec-Apr | 30,30,30,30,10 | 25,30,30,15,5 | 20,20,20,15 | 25,25,25,30,25 |
| Percentage retained | 20.6 | 18.1 | 29.9 | 24.2 |

**Fig. 6.** Table S2. A new table in response to reviewer comments

| Parameter Name | Unconstrained | CCN$_{0.2}$ | CCN$_{1.0}$ | Non-sea-salt sulfate | N$_{700}$ | Combined |
|---|---|---|---|---|---|---|
| BL_Nuc | 0.1,10.0 [0.3,3.2] | 0.1,10.0 [0.3,**3.5**] | 0.1,10.0 [0.3,3.0] | 0.1,10.0 [0.3,3.3] | 0.1,10.0 [0.3,3.2] | 0.1,10.0 [0.3,3.5] |
| Ageing | 0.3,10.0 [2.7,7.6] | 0.3,10.0 [3.0,7.9] | 0.3,10.0 [2.5,7.5] | 0.3,10.0 [2.7,7.6] | 0.3,10.0 [2.6,7.5] | 0.3,10.0 [2.7,7.6] |
| Acc_Width | 1.2,1.8 [1.4,1.6] | 1.2,1.8 [**1.3**,1.7] | 1.2,1.8 [1.4,1.7] | 1.2,1.8 [1.4,1.7] | 1.2,1.8 [**1.3**,1.7] | 1.2,1.8 [1.3,1.7] |
| Ait_Width | 1.2,1.8 [1.3,1.6] | 1.2,1.8 [1.3,1.7] | 1.2,1.8 [1.3,1.6] | 1.2,1.8 [1.3,1.7] | 1.2,1.8 [1.3,1.7] | 1.2,1.8 [1.3,1.6] |
| Cloud_pH | 4.6,7.0 [5.2,6.4] | 4.6,7.0 [**5.1**,6.4] | 4.6,7.0 [**5.1**,**6.2**] | 4.6,7.0 [5.2,6.4] | 4.6,7.0 [5.2,6.4] | 4.6,7.0 [5.1,6.2] |
| Carb_FF_Ems | 0.5,2.0 [0.7,1.4] | 0.5,2.0 [0.7,1.4] | 0.5,2.0 [0.7,1.4] | 0.5,2.0 [0.7,1.4] | 0.5,2.0 [0.7,1.4] | 0.5,2.0 [0.7,1.4] |
| Carb_BB_Ems | 0.25,4.00 [0.50,2.00] | 0.25,4.00 [0.52,**2.16**] | 0.25,4.00 [**0.48**,2.01] | 0.25,4.00 [0.50,2.01] | 0.25,4.00 [0.49,2.03] | 0.25,4.00 [0.49,2.06] |
| Carb_Res_Ems | 0.25,4.00 [0.50,2.00] | 0.25,4.00 [**0.45**,**1.78**] | 0.25,4.00 [0.48,2.02] | 0.25,4.00 [0.49,2.00] | 0.25,4.00 [0.50,2.02] | 0.25,4.00 [0.48,1.94] |
| Carb_FF_Diam | 30,90 [45,75] | 30,90 [45,**76**] | 30,90 [44,75] | 30,90 [45,75] | 30,90 [45,75] | 30,90 [45,76] |
| Carb_BB_Diam | 90,300 [143,248] | 90,300 [141,**250**] | 90,300 [**140**,249] | 90,300 [142,248] | 90,300 [141,248] | 90,300 [141,249] |
| Carb_Res_Diam | 90,500 [193,398] | 90,500 [193,**404**] | 90,500 [**190**,399] | 90,500 [192,400] | 90,500 [193,400] | 90,500 [189,400] |
| Prim_SO4_Frac | 1.0e-6,1.0e-1 [1.8e-5,5.6e-3] | 1.0e-6,1.0e-1 [1.7e-5,5.6e-3] | 1.0e-6,1.0e-1 [**1.3e-5**,**4.2e-3**] | 1.0e-6,1.0e-1 [1.7e-5,5.6e-3] | 1.0e-6,1.0e-1 [1.6e-5,6.0e-3] | 1.0e-6,1.0e-1 [1.6e-5,5.2e-3] |
| Prim_SO4_Diam | 3,100 [27,76] | 3,100 [26,75] | 3,100 [**29**,**78**] | 3,100 [27,76] | 3,100 [26,77] | 3,100 [28,77] |
| Sea_Spray | 0.1,8.0 [0.4,2.8] | 1.5,8.0 [2.7,3.8] | **1.9**,8.0 [**3.8**,**5.7**] | 0.1,8.0 [0.3,2.8] | 1.5,**5.2** [2.5,3.6] | 1.6,5.1 [2.6,3.7] |
| Anth_SO2 | 0.6,1.5 [0.8,1.2] | 0.6,1.5 [0.8,1.2] | 0.6,1.5 [0.7,1.2] | 0.6,1.5 [0.8,1.2] | 0.6,1.5 [0.8,1.2] | 0.6,1.5 [0.8,1.2] |
| Volc_SO2 | 0.7,2.4 [1.0,1.8] | 0.7,2.4 [1.0,1.8] | 0.7,2.4 [1.0,1.8] | 0.7,2.4 [1.0,1.8] | 0.7,2.4 [1.0,1.8] | 0.7,2.4 [1.0,1.8] |
| BVOC_SOA | 0.8,5.4 [1.3,3.4] | 0.8,5.4 [1.3,3.5] | 0.8,5.4 [1.4,3.5] | 0.8,5.4 [1.3,3.4] | 0.8,5.4 [1.3,3.4] | 0.8,5.4 [1.3,3.4] |
| DMS | 0.5,2.0 [0.7,1.4] | 0.5,2.0 [0.7,1.5] | 0.5,2.0 [0.7,1.4] | 0.5,2.0 [**0.8**,1.5] | 0.5,2.0 [0.7,1.4] | 0.5,2.0 [0.8,1.3] |
| Dry_Dep_Ait | 0.5,2.0 [0.7,1.4] | 0.5,2.0 [0.7,1.4] | 0.5,2.0 [0.7,1.3] | 0.5,2.0 [0.7,1.4] | 0.5,2.0 [0.7,1.4] | 0.5,2.0 [0.7,1.4] |
| Dry_Dep_Acc | 0.1,10.0 [0.3,3.2] | 0.1,9.3 [**0.2**,**0.9**] | 0.1,**6.7** [**0.2**,1.0] | 0.1,10.0 [0.3,1.9] | 0.1,10.0 [0.3,3.2] | 0.1,6.4 [0.2,0.8] |
| Dry_Dep_SO2 | 0.2,5.0 [0.4,2.2] | 0.2,5.0 [0.4,2.2] | 0.2,5.0 [0.4,2.4] | 0.2,5.0 [0.4,2.2] | 0.2,5.0 [0.4,2.2] | 0.2,5.0 [0.4,2.2] |
| Kappa_OC | 0.1,0.6 [0.2,0.5] | 0.1,0.6 [0.2,0.5] | 0.1,0.6 [0.2,0.5] | 0.1,0.6 [0.2,0.5] | 0.1,0.6 [0.2,0.5] | 0.1,0.6 [0.2,0.5] |
| Sig_W | 0.1,0.7 [0.3,0.5] | 0.1,0.7 [0.2,0.6] | 0.1,0.7 [0.2,0.6] | 0.1,0.7 [0.2,0.6] | 0.1,0.7 [0.2,0.6] | 0.1,0.7 [0.2,0.6] |
| Dust | 0.5,2.0 [0.7,1.4] | 0.5,2.0 [0.7,1.4] | 0.5,2.0 [0.7,1.4] | 0.5,2.0 [0.7,1.4] | 0.5,2.0 [0.7,1.4] | 0.5,2.0 [0.7,1.4] |
| Rain_Frac | 0.3,0.7 [0.4,0.6] | 0.3,0.7 [0.4,0.6] | 0.3,0.7 [0.4,0.6] | 0.3,0.7 [0.4,0.6] | 0.3,0.7 [0.4,0.6] | 0.3,0.7 [0.4,0.6] |
| Cloud_Ice_Thresh | 0.1,0.5 [0.2,0.4] | 0.1,0.5 [0.2,**0.3**] | 0.1,0.5 [0.2,0.4] | 0.1,0.5 [0.2,0.4] | 0.1,0.5 [0.2,0.4] | 0.1,0.5 [0.2,0.4] |

**Fig. 7.** Table S3. A new table in response to reviewer comments